# Unifying Homophily and Heterophily for Spectral Graph Neural Networks via Triple Filter Ensembles

**Rui Duan[1], Mingjian Guang[2], Junli Wang[3], Chungang Yan[3], Hongda Qi[4],**

**Wenkang Su[1]\*, Can Tian[1]\*, Haoran Yang[3]**

[1]School of Computer Science and Cyber Engineering, Guangzhou University, China,
[2]Donghua University, China, [3]Tongji University, China [4]Shanghai Normal University, China
[1]{duan, swk1004, tiancan}@gzhu.edu.cn; [2]guangmingjian@dhu.edu.cn;
[3]{junliwang, yanchungang, 2010498}@tongji.edu.cn; [4]hongda_qi@shnu.edu.cn

## Abstract

Polynomial-based learnable spectral graph neural networks (GNNs) utilize polynomial to approximate graph convolutions and have achieved impressive performance on graphs. Nevertheless, there are three progressive problems to be solved. Some models use polynomials with better approximation for approximating filters, yet perform worse on real-world graphs. Carefully crafted graph learning methods, sophisticated polynomial approximations, and refined coefficient constraints leaded to overfitting, which diminishes the generalization of the models. How to design a model that retains the ability of polynomial-based spectral GNNs to approximate filters while it possesses higher generalization and performance? In this paper, we propose a spectral GNN with **t**riple **f**ilter **e**nsemble (TFE-GNN), which extracts homophily and heterophily from graphs with different levels of homophily adaptively while utilizing the initial features. Specifically, the first and second ensembles are combinations of a set of base low-pass and high-pass filters, respectively, after which the third ensemble combines them with two learnable coefficients and yield a graph convolution (TFE-Conv). Theoretical analysis shows that the approximation ability of TFE-GNN is consistent with that of ChebNet under certain conditions, namely it can learn arbitrary filters. TFE-GNN can be viewed as a reasonable combination of two unfolded and integrated excellent spectral GNNs, which motivates it to perform well. Experiments show that TFE-GNN achieves high generalization and new state-of-the-art performance on various real-world datasets. The source code of GEN is publicly available at https://github.com/graphNN/TFEGNN

## 1 Introduction

Graph neural networks (GNNs) are competitive in graph-related tasks (Scarselli et al., 2008; Yang et al., 2023; Shirzad et al., 2023; Duan et al., 2024) and can be divided into two main categories: spatial-based (Xu et al., 2019) and spectral-based (Shirzad et al., 2023; Tao et al., 2023; He et al., 2022; Bo et al., 2023b; Guo et al., 2023) GNNs. Spectral graph convolutions in the spectral domain of the graph Laplace matrix, i.e., the spectral graph filters, are the core component of spectral-based GNNs. We further classify spectral-based GNNs into two categories based on whether their graph convolutions can be learned or not.

The first class of spectral-based GNNs' graph convolutions are predetermined, i.e., they filter the graph signals (features) in a fixed way. Graph convolutional networks (GCNs) (Kipf & Welling, 2016)

---

*Corresponding authors

38th Conference on Neural Information Processing Systems (NeurIPS 2024).

and their variants (Rong et al., 2020; Chen et al., 2020; Yang et al., 2021) utilize only the first two Chebyshev polynomials to simplify ChebNet (Defferrard et al., 2016), and their graph convolutions are the fixed low-pass filters.

The second class of spectral-based GNNs' graph convolutions are learnable, i.e., their filters are variable and they filter the graph signals in learnable way. ChebNet (Defferrard et al., 2016) utilizes Chebyshev polynomials to approximate the graph convolutions and it can learn arbitrary filters in theory (Balcilar et al., 2021; He et al., 2022). CayleyNet (Levie et al., 2018) utilizes Cayley polynomials to learn the graph convolutions, and ARMA (Bianchi et al., 2021a) learns the rational graph convolutions by using the Auto-Regressive Moving Average filters family (Narang et al., 2013). GPR-GNN (Chien et al., 2021) and BernNet (He et al., 2021a) use Monomial and Bernstein polynomials to approximate the graph convolutions. ChebNetII (He et al., 2022) revisits ChebNet and makes learned coefficients more legal through Chebyshev interpolation. EvenNet (Lei et al., 2022) ignores odd-hop neighbors and improves the robustness of GNNs by using the even-polynomial graph filter. PCNet (Li et al., 2023) uses the Possion-Charlier polynomials to approximate the graph filter and constrain the coefficients. Just like EvenNet, the heterophilic graph heat kernel provided by PCNet pushes odd-hop neighbors away and aggregates even-hop neighbors. FavardGNN (Guo & Wei, 2023) learns a polynomial basis from the space of possible orthonormal bases and OptBasisGNN (Guo & Wei, 2023) computes the optimal basis for a given graph structure and signal. Specformer (Bo et al., 2023a) encodes the set of eigenvalues and performs self-attention in the spectral domain, which leads to a learnable set-to-set spectral filter.

Despite polynomial-based learnable spectral GNNs have achieved impressive performance on graphs, there are three progressive problems to be solved. *First, some polynomial-based models use polynomials with better approximation than some other models when approximating filters, but the former's performance is lagging behind that of the latter on real-world graphs.* For example, GPR-GNN and BernNet outperform ChebNet, even though they use polynomials that are weaker than Chebyshev polynomials in approximation theory (He et al., 2022). ChebNetII, an enhanced version of ChebNet, whose performance still lags behind that of PCNet using Possion-Charlier polynomials. The important factors influencing the real-world performance of such GNNs are graph learning methods, polynomial approximation and coefficient constraints, but of course there are others as well, in any case not only the polynomial approximation ability. The following two facts exist: ChebNetII outperforms GPR-GNN and BernNet through refined coefficient constraints; and PCNet outperforms ChebNetII through the carefully crafted graph learning method, i.e., pushing odd-hop neighbors away to match the structural properties of heterophilic graph.

The second problem was raised based on the answer to the first problem. *Carefully crafted graph learning methods, sophisticated polynomial approximations, and refined coefficient constraints leaded to overfitting while improving models' performance, which diminishes the generalization of the models.* FFKSF (Zeng et al., 2023) attributes the degradation of polynomial filters' performance to the overfitting of polynomial coefficients. ChebNetII (He et al., 2022) further constrains the coefficients to enable them easier to be optimized. ARMA (Bianchi et al., 2021b) suggests that the filter will overfit the training data when aggregating high-order neighbor information. Whereas the order of polynomial-based spectral GNNs is usually large to increase the approximation of the polynomials, which directs them to obtain high-order neighborhood information, and then leads to overfitting. Therefore, it is reasonable to assume that carefully crafted graph learning methods, sophisticated polynomial approximations, and refined coefficient constraints lead to overfitting of the models, which diminishes generalization of the models. Desired learnable spectral GNNs can learn various graph convolutions, which motivates them to extract homophily from heterophilic graph and vice versa. Instead of ignoring odd-hop or even-hop neighbors, which makes the model miss important neighbor information.

Finally, the third problem was raised. *How to design a model that retains the ability of polynomial-based spectral GNNs to approximate filters while it possesses higher generalization and performance?* In this paper, inspired by ensemble learning (Schapire, 1990; Hansen & Salamon, 1990; Zhou, 2012) (see in section 3.1), we design **t**riple **f**ilter **e**nsemble (TFE) mechanism to adaptively extract homophily and heterophily from graphs with different levels of homophily while utilizing the initial features, where the first and second ensembles combine a set of base low-pass and high-pass graph filter, respectively, and the third ensemble combines them by two learnable coefficients. The third ensemble of TFE will yield a graph convolution (TFE-Conv) used to filter the graph signal. The filtered signal is

fed into a fully connected linear neural network (NN), whose output is then passed through a *softmax* layer to obtain the prediction.

TFE-GNN does not impose refined constraints on the coefficients and does not design very complex learning methods, which possesses higher generalization. The key difference between TFE-GNN and prior models is that TFE-GNN retains the ability of polynomial-based spectral GNNs while getting rid of polynomial computations, coefficient constraints, and specific scenarios. We describe the differences between TFE-GNN and several other recent methods (Li et al., 2024; Huang et al., 2024b,a) in Appendix B.7. TFE-GNN also offers the following three advantages. We theoretically demonstrate that the approximation ability of TFE-GNN agrees with that of ChebNet under certain conditions, as outlined in Theorem 1, i.e., can learn arbitrary filters. Theorem 2 shows that TFE-GNN is a reasonable combination of two excellent polynomial-based spectral GNNs, which motivates it to perform well. TFE extract the initial information, homophily and heterophily from graphs adaptively, which allows TFE-GNN to be applied to various homophily level cases. Experiments show that TFE-GNN achieves new state-of-the-art performance on datasets, and the homophily levels measured by the edge homophily ratio (Zhu et al., 2020) for these datasets is 0.06, 0.21, 0.23, 0.30, 0.57, 0.74, 0.80, 0.81, and 0.93.

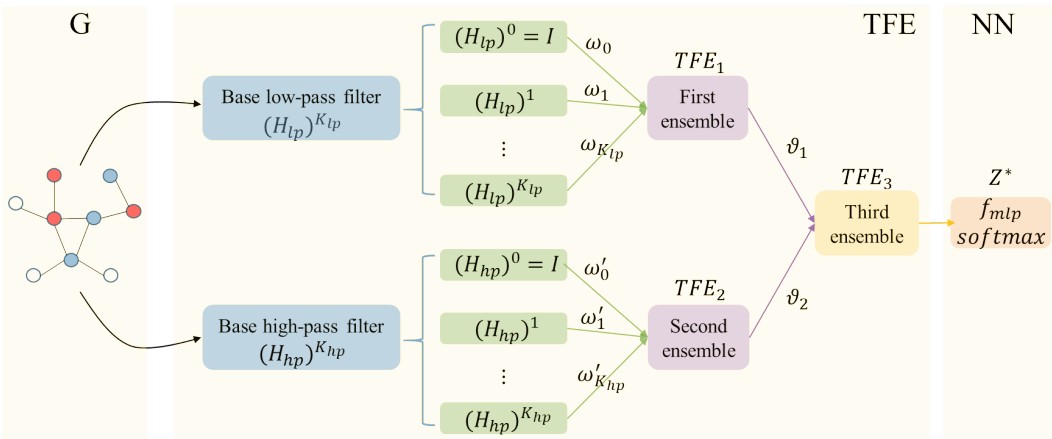

Figure 1: An illustration of TFE-GNN.

## 2 Preliminaries

**Notations.** Given a graph $G = (E, V)$ with node set $V$ and edge set $E \subseteq V \times V$. Let $n = |V|$ denote the size of the node set, i.e., the number of nodes. This paper uses $x \in \mathbb{R}^n$ to denote the graph signals, and $x(i)$ to denote the signal at node $i$. $Y \in \{0, 1\}^{n \times C}$ denotes label matrix of $G$, and $Y_i$ is the label vector of node $i$, where $C$ is the number of classes. We denote $X \in \mathbb{R}^{n \times d_0}$ as the initial feature matrix of $G$, denote the adjacency matrix of $G$ as $A \in \{0, 1\}^{n \times n}$, and denotes the degree matrix as $D$, where $D_{ii} = \sum_{j=0}^{n} A_{ij}$, and $d_0$ is the initial feature dimension. The (combinatorial) graph Laplacian is defined as $L = D - A$, and its eigendecomposition is $L = U \Lambda U^T$. The columns $u_i$ of $U \in \mathbb{R}^{n \times n}$ are orthonormal eigenvectors, namely the graph Fourier basis, and $\Lambda = diag([\lambda_1, ..., \lambda_n])$ is the diagonal matrix of eigenvalues. We also call these eigenvalues frequencies.

### 2.1 Metrics of Homophily

The homophily metrics are used to define the homophily level of a graph by considering the different relationships between node labels/features and graph structures, including edge homophily (Zhu et al., 2020), node homophily (Pei et al., 2020), class homophily (Lim et al., 2021b), etc. In this paper, we use the edge homophily ratio $ehr$ to measure the ratio of intra-class edges contained in a graph as the homophily level:

$$ehr = \frac{|(u, v) : (u, v) \in E \wedge Y_u = Y_v|}{|E|}, \tag{1}$$

where $u, v$ are nodes, $Y_u$ is the label vector of $u$, and $Y_v$ is the label vector of $v$. The value of $ehr$ close to 1 corresponds to strong homophily, while the value of $ehr$ close to 0 indicates strong heterophily.

## 2.2 Graph Filter

The graph filters are core components of spectral GNNs. We classify the graph filters into three categories based on the type of signal they filter: low-pass, high-pass and full-pass filters. The high-pass filter $H_{hp}$ are more suitable for extracting high-frequency signals (Bo et al., 2023a; Yang et al., 2022). Empirically, the commonly used high-pass filters are the symmetric normalized Laplacian $L_{sym} = D^{-1/2}LD^{-1/2} = I - D^{-1/2}AD^{-1/2}$ and the random walk normalized Laplacian $L_{rw} = D^{-1}L = I - D^{-1}A$. The low-pass filter $H_{lp}$ are more suitable for extracting low-frequency signals and it is the affinity (transition) matrix of $H_{hp}$, i.e., $H_{lp}$ corresponding to $H_{hp}$ above are $L_{sym}^a = I - L_{sym} = D^{-1/2}AD^{-1/2}$ and $L_{rw}^a = I - L_{rw} = D^{-1}A$. The full-pass filter $H_{fp}$ is the identity matrix $I$ and retains all graph initial signals.

## 2.3 Learnable Spectral GNNs

Spectral-based GNNs create the spectral graph convolutions (filters) in the domain of Laplacian spectrum and many methods use the polynomial spectral filters to achieve graph convolutions, such as ChebNet Defferrard et al. (2016), GPR-GNN Chien et al. (2021), BernNet He et al. (2021a), ChebNetII He et al. (2022), PCNet (Li et al., 2023), FavardGNN (Guo & Wei, 2023), etc. We begin by describing how the signal $x$ is filtered by $h$:

$$y = h(L)x = h(U\Lambda U^T)x = Uh(\Lambda)U^T x = U diag([h(\lambda_1), ..., h(\lambda_n)])U^T x, \quad (2)$$

where $y$ denotes the filtering results of $x$, and $h$ denotes the spectral filter, which is a function on the eigenvalues of the graph Laplacian matrix $L$. Existing studies have replaced nonparametric filters with polynomial filters:

$$h(\Lambda) = \sum_{k=0}^{K-1} \theta_k \Lambda^k, \quad (3)$$

where $\theta_k \in \mathbb{R}^K$ is a vector of polynomial coefficients. We bring Equation 3 into Equation 2:

$$y = U \sum_{k=0}^{K-1} \theta_k \Lambda^k U^T x = \sum_{k=0}^{K-1} \theta_k U\Lambda^k U^T x = \sum_{k=0}^{K-1} \theta_k L^k x, \quad (4)$$

We describe polynomial-based spectral GNNs in terms of ChebNet (Defferrard et al., 2016) and its enhanced version ChebNetII (He et al., 2022), including how they approximate graph convolution with polynomials and how constraining the coefficients. Other related methods are similar, such as EvenNet (Lei et al., 2022) and PCNet (Li et al., 2023). ChebNet uses Chebyshev polynomial to approximate the filtering operation, which is a remarkable attempt:

$$y = \sum_{k=0}^{K-1} \theta_k T_k(\tilde{L})x, \quad (5)$$

where $\tilde{L} = 2L_{sym}/\lambda_{max} - I$ denotes the scaled graph Laplacian matrix, $\lambda_{max}$ is the largest eigenvalue of $L$ and $\theta$ is a vector of Chebyshev coefficients. Chebyshev polynomial $T_k(\boldsymbol{x})$ of order $k$ can be recursively defined as $T_k(\boldsymbol{x}) = 2\boldsymbol{x}T_{k-1}(\boldsymbol{x}) - T_{k-2}(\boldsymbol{x})$ with $T_0(\boldsymbol{x}) = 1$ and $T_1(\boldsymbol{x}) = \boldsymbol{x}$. ChebNet's structure is:

$$\hat{Y} = \sum_{k=0}^{K-1} T_k(\tilde{L})XW_k, \quad (6)$$

where $W_k$ are the trainable weights, which contain the Chebyshev coefficients $\theta_k$. ChebNetII (He et al., 2022) proposes ChebBase to determine who is more competitive, Chebyshev basis or other bases.

$$\hat{Y} = \sum_{k=0}^{K} \theta_k T_k(\tilde{L})f(X), \quad (7)$$

where $f(X)$ is a Multi-Layer Perceptron (MLP). ChebNetII believes that ChebNet learns the illegal coefficient by analyzing a series of polynomial filters. Therefore, it then proposes ChebBase/$k$, which is an improvement on ChebNet and constrains the coefficients with $\theta_k/k$ (Equation 7). Finally, ChebNetII is proposed.

$$\hat{Y} = \frac{2}{K+1} \sum_{k=0}^{K} \sum_{j=0}^{K} \gamma_j T_k(x_j) T_k(\tilde{L}) f(X), \tag{8}$$

where $\gamma_j$ is the learnable coefficient, and it links Equation 7 and Equation 8 with $\theta_k = \frac{2}{K+1} \sum_{j=0}^{K} \gamma_j T_k(x_j)$, i.e., the learnable coefficients constrain.

## 3 Methodology

We propose a spectral GNN with triple filter ensemble (TFE-GNN) to solve three progressive problems in polynomial-based learnable spectral GNNs. In this section, we describe the methodology of TFE-GNN in detail and theoretically prove its approximation capabilities, including motivations, TFE-Conv, TFE-GNN, time complexity and scalability and theoretical analysis.

### 3.1 Motivations

Polynomial-based learnable spectral GNNs (He et al., 2022; Li et al., 2023; Guo & Wei, 2023) performs well on homophilic and heterophilic graphs, because their graph convolutions are flexible and variable. However, carefully crafted graph learning methods, sophisticated polynomial approximations, and refined coefficient constraints leaded to overfitting, which diminishes GNNs' generalization.

Inspired by the following properties of ensemble learning (Schapire, 1990; Hansen & Salamon, 1990; Zhou, 2012): the strong classifier determined by the base classifiers can be more accurate than any of them if the base classifiers are accurate and diverse; and this strong classifier retains the characteristics of the base classifier to some extent. First, we combine a set of weak base low-pass filter to determine a strong low-pass filter that can extract homophily. Then, we use the same method to extract heterophily. Finally, TFE-Conv is generated by combining the above two strong filters with two learnable coefficients, which retains the characteristics of both two strong filters, i.e., it can extract homophily and heterophily from graphs adaptively. TFE-Conv and TFE-GNN are shown in Figure 1.

### 3.2 TFE-Conv

We design triple filter ensemble (TFE) mechanism for combining low-pass and high-pass filters to yield a graph convolution (TFE-Conv), which can match various graph structures adaptively without carefully crafted graph learning methods, i.e., can extract homophily and heterophily adaptively from graphs with different homophily level. The first ensemble of TFE is formalized as follows:

$$TFE_1 = EM_1\{\omega_0 I, \omega_1 H_{lp}, \omega_2(H_{lp})^2, \cdots, \omega_{K_{lp}}(H_{lp})^{K_{lp}}\}, \tag{9}$$

where $TFE_1$ denotes the result of combining a set of base low-pass filters, which is a graph convolution that can extract homophily while utilizing the initial signals, $EM_1$ denotes the ensemble method of the first ensemble, $EM_1\{h_0, h_1, \cdots, h_K\}$ denotes the combination of elements $h_0, h_1, \cdots, h_K$ with $EM_1$, $\omega$ are the learnable coefficients, and $K_{lp}$ is the order of the first ensemble. The second ensemble is similar to the first ensemble and is formalized as follows:

$$TFE_2 = EM_2\{\omega_0' I, \omega_1' H_{hp}, \omega_2'(H_{hp})^2, \cdots, \omega_{K_{hp}}'(H_{hp})^{K_{hp}}\}, \tag{10}$$

where $TFE_2$ denotes the result of the second ensemble, which can extract heterophily while utilizing the initial signals, $EM_2$ denotes the ensemble method of the second ensemble, $\omega'$ are the learnable coefficients, and $K_{lp}$ is the order of the second ensemble. The third ensemble combines $TFE_1$ and $TFE_2$ with two learnable coefficients $\vartheta_1$ and $\vartheta_2$:

$$TFE_3 = EM_3\{\vartheta_1 TFE_1, \vartheta_2 TFE_2\}, \tag{11}$$

where $TFE_3$ denotes the result of the third ensemble, i.e., TFE-Conv, and $EM_3$ denotes the ensemble method of the third ensemble. The learnable coefficients $\vartheta_1$ and $\vartheta_2$ used to combine the two strong graph convolutions $TFE_1$ and $TFE_2$ guarantee $TFE_3$'s adaptivity.

### 3.3 TFE-GNN

TFE-Conv matches various graph structures adaptively, which facilitates the filtering of the graph signals (features) $X$. The filtered signal is fed into an MLP, whose output is then passed through a *softmax* layer for obtaining the prediction. This is the forward propagation of TFE-GNN and is coupled with the cross-entropy loss and the backpropagation mechanism to form the complete TFE-GNN. The formalization of $X$ being filtered by TFE-Conv is:

$$Z = TFE_3 \cdot X = EM_3\{\vartheta_1 TFE_1, \vartheta_2 TFE_2\} \cdot X = EM_3\{\vartheta_1 TFE_1 \cdot X, \vartheta_2 TFE_2 \cdot X\}, \quad (12)$$

where $TFE_1$ and $TFE_2$ are the graph convolutions obtained from the first and second ensembles, respectively, which are similar to $TFE_3$ in that they filter the graphical signal as follows: $Z_{lp} = TFE_1 \cdot X$ and $Z_{hp} = TFE_2 \cdot X$. Thus, Equation 12 can be rewritten in the following form:

$$Z = EM_3\{\vartheta_1 Z_{lp}, \vartheta_2 Z_{hp}\}. \quad (13)$$

We decouple transformation/prediction and feature propagation (Rong et al., 2020; He et al., 2022) for TFE-GNN and formalize TFE-GNN used for node classification:

$$\begin{aligned}
\tilde{Z} &= f_{mlp}(Z) \\
Z^* &= softmax(\tilde{Z}) \\
\mathscr{L} &= -\sum_{r \in \mathbb{Y}_{\mathbb{L}}} Y_r^\top log(Z_r^*),
\end{aligned} \quad (14)$$

where $\mathscr{L}$ denotes the cross-entropy loss, $\mathbb{Y}_{\mathbb{L}}$ denotes the training set with labels, and $\top$ denotes the vector transpose.

### 3.4 Time Complexity and Scalability

Similar to some spectral GNNs (He et al., 2022; Li et al., 2023), the triple filter ensemble mechanism does not influence the time complexity magnitude of the TFE-GNN, i.e., the time complexity of TFE-GNN is linear to $K_{lp}+K_{hp}$ when $EM_1$, $EM_2$, and $EM_3$ take summation. Specifically, the time complexity of message propagation is $O((K_{lp} + K_{hp})|E|C)$, the time complexity of the combination of $H_{gf}$ with respectively $\omega$ and $\omega'$ (Equations 9 and 10) is $O((K_{lp} + K_{hp})nC)$, and the time complexity of the coefficient calculation is not greater than $O(K_{lp} + K_{hp})$. We report the training time overhead of the different spectral GNNs in Appendix B.6.

We scale TFE-GNN by exchanging the order of message propagation and feature dimensionality reduction: $\widetilde{Z} = EM_3\{\vartheta_1 Z_{lp}, \vartheta_2 Z_{hp}\} = EM_3\{\vartheta_1 TFE_1 f_{mlp}(X), \vartheta_2 TFE_2 f_{mlp}(X)\}$. We use a sparse form of the adjacency matrix of large graphs, which greatly reduces the space required for TFE-GNN. Therefore, TFE-GNN scales well to large graphs and high-dimensional feature spaces.

### 3.5 Theoretical Analysis

ChebNet has been shown to learn arbitrary filters in theory (Balcilar et al., 2021; He et al., 2022). We find a connection between TFE-GNN and ChebNet and then analyze the conditions under which they transform into each other. We prove that TFE-GNN is equal to ChebNet under certain conditions, i.e., they have the same approximation ability, so TFE-GNN can also learn arbitrary filters.

**Theorem 1.** *TFE-GNN and ChebNet can be transformed into each other under the following conditions, (1) learning the proper coefficients $\omega$, $\omega'$, $\vartheta$ and ChebNet' coefficient $\theta$, (2) the ensemble methods $EM_1$, $EM_2$ and $EM_3$ take **summation**, the base high-pass filter $H_{hp}$ takes the symmetric normalized Laplacian $L_{sym}$ and the base low-pass filter $H_{lp}$ takes the affinity (transition) matrix of $L_{sym}$, and (3) $K_{hp} = K_{lp} = K - 1$. Thus, TFE-GNN can also learn arbitrary filters.*

Theorem 1 (proof in Appendix A.1) shows that TFE-GNN matches various graph structures adaptively while learning arbitrary filters under certain conditions. The proof of Theorem 1 shows that ChebNet can be unfolded into a combination of high-pass or low-pass filters, or a combination of high-pass and low-pass. TFE-GNN, on the other hand, is a combination of two different unfoldings of ChebNet. Condition (1) of Theorem 1 is shown in its proof, the $EM_1$ and $EM_2$ in condition (2) are taken to **summation** to correspond to the expansion of the Chebyshev polynomials, while $EM_3$ is an ensemble method capable of preserving the properties of the model, such as **summation** and **concatenation**, and condition (3) motivates the agreement between the orders of TFE-GNN and ChebNet.

Polynomial-based spectral graph GNNs can all be unfolded into combinations of filters, which is similar to the idea of filter ensemble in TFE-GNN. TFE-GNN is a combination of two polynomial-based GNNs under certain conditions by displacing its base filters and setting $K_{lp}$ and $K_{hp}$.

**Theorem 2.** *TFE-GNN can be rewritten in the following form, with certain conditions to be satisfied, which is a combination of two polynomial-based learnable spectral GNNs:* $Z^* = softmax(f_{mlp}(\vartheta_1 \sum_{k=0}^{K'} \bar{\theta}_k^1 P_k^1 (\bar{H}_{gf}^1)^k X \bigoplus \vartheta_2 \sum_{k=0}^{K''} \bar{\theta}_k^2 P_k^2 (\bar{H}_{gf}^2)^k X))$, *where $P_k$ denote polynomials used for approximation, $\bar{\theta}$ are the learnable coefficients, $\bar{H}_{gf}$ denote graph filters, and $\bigoplus$ denotes $EM_3$. Conditions are (1) learning the proper coefficients $\omega$, $\omega'$, $\vartheta$, $\bar{\theta}^1$, and $\bar{\theta}^2$, (2) the ensemble methods $EM_1$, $EM_2$ take **summation** and $EM_3$ takes ensemble method capable of preserving the properties of the model, such as **summation** and **concatenation**, the base high-pass filter $H_{hp}$ takes $\bar{H}_{gf}^2$ and the base low-pass filter $H_{lp}$ takes $\bar{H}_{gf}^1$, and (3) $K_{lp} = K'$ and $K_{hp} = K''$. Thus, TFE-GNN can match various graph structures adaptively.*

The proof of Theorem 2 is reported in Appendix A.1. Theorems 1 and 2 show that polynomial-based learnable spectral GNNs are able to learn different filters, and thus perform well on both homophilic and heterophilic graphs. In contrast, fixed-filter GNNs can only filter graph signals based on their filter forms. TFE-GNN combines different filters directly, preserving the ability of the different filters while reducing overfitting problem. Theorem 2 also shows that TFE-GNN will perform well on real-world datasets: it is a reasonable combination of two excellent polynomial-based spectral GNNs. We discuss the **limitations** of TFE-GNN in Appendix A.2.

## 4 Experiments

In this section, we conduct experiments to evaluate the proposed TFE-GNN against the state-of-the-art (SOTA) GNNs on real-world datasets and conduct ablation study, generalization, visualization (Appendix B.3), hyper-parameters (Appendix B.5), and time efficiency (Appendix B.6) analysis to verify TFE-GNN's excellent.

**Datasets and experimental setup.** We evaluate TFE-GNN on several real-world datasets for supervised node classification and chose 11 graphs with various levels of homophily, including 4 citation graphs (Kipf & Welling, 2016) Cora, Citeseer, Pubmed, and Cora-Full (Bojchevski & Günnemann, 2017), 2 Co-authorship graphs (Shchur et al., 2018) Coauthor CS and Coauthor Physics, 2 Wikipedia graphs (Rozemberczki et al., 2021) Chameleon and Squirrel, and 3 WebKB graphs (Pei et al., 2020) Texas, Cornell, and Wisconsin. The dataset statistics are summarized in Table 1. In addition, we choose four additional datasets, i.e. roman-empire (Platonov et al., 2023), amazon-rating (Platonov et al., 2023), fb100-Penn94 (Lim et al., 2021a) and genius (Lim et al., 2021a), to further validate the classification performance, generalization and scalability of TFEGNN. Detailed dataset statistics and experimental results are reported in the Appendix B.1. All experiments are carried out on the machine with Linux system, two NVIDIA Tesla V100 and twelve Intel(R) Xeon(R) Gold 5220 CPU @2.20GHz.

Table 1: Dataset statistics.

| DATASETS | CORA | CITESEER | PUBMED | CORA-FULL | CS | PHYSICS | CHAMELEON | SQUIRREL | WISCONSIN | TEXAS | CORNELL |
|---|---|---|---|---|---|---|---|---|---|---|---|
| NODES | 2708 | 3327 | 19717 | 19793 | 18333 | 34493 | 2277 | 5201 | 251 | 183 | 183 |
| EDGES | 10556 | 9104 | 88648 | 62421 | 81894 | 247962 | 36051 | 216933 | 466 | 309 | 295 |
| FEATURES | 1433 | 3703 | 500 | 8710 | 6805 | 8415 | 1703 | 2089 | 931 | 1703 | 1703 |
| CLASSES | 7 | 6 | 3 | 70 | 15 | 5 | 5 | 5 | 5 | 5 | 5 |
| *ehr* | 0.81 | 0.74 | 0.80 | 0.57 | 0.81 | 0.93 | 0.23 | 0.22 | 0.21 | 0.06 | 0.30 |

**TFE-GNN Settings.** There are many options available for $H_{lp}$, $H_{hp}$, $EM_1$, $EM_2$, and $EM_3$ in TFE-GNN, and we choose common and frequently used options for them to make a broad and fair comparison between TFE-GNN and other SOTA GNNs. $H_{hp}$ and $H_{lp}$ take the symmetric normalized Laplacian $L_{sym}$ and $L_{sym}^a$ respectively. We add self-loops to the graph in practice, so $L_{sym} = I - \tilde{D}^{-1/2}\tilde{A}\tilde{D}^{-1/2}$ and $L_{sym}^a = \tilde{D}^{-1/2}\tilde{A}\tilde{D}^{-1/2}$, where $\tilde{A} = A + I$, and $\tilde{D}_{ii} = \sum_{j=0}^n \tilde{A}_{ij}$. We use generalized normalization about $L_{sym}$ to alleviate the overcorrelation issue in spectral GNNs (Yang et al., 2022; Li et al., 2023) and keep $L_{sym}^a$ unchanged, namely $L_{sym} = I - \tilde{D}^{-\eta}\tilde{A}\tilde{D}^{-\eta}$. $EM_1$ and $EM_2$ take summation, and $EM_3$ takes summation or concatenation. Thus, TFE-Conv

Table 2: Mean accuracy of different models on datasets for full-supervised node classification.

| DATASETS | CORA | CITESEER | PUBMED | CS | PHYSICS |
|---|---|---|---|---|---|
| *ehr* | 0.81 | 0.74 | 0.80 | 0.81 | 0.93 |
| MLP | 76.89±0.97 | 76.52±0.89 | 86.14±0.25 | 94.76±0.51 | 96.52±0.66 |
| GCNs | 87.18±1.12 | 79.85±0.78 | 86.79±0.31 | 93.11±0.19 | 96.66±0.74 |
| ARMA | 87.13±0.80 | 80.04±0.55 | 86.93±0.24 | 92.14±0.35 | 95.11±0.19 |
| APPNP | 88.16±0.74 | 80.47±0.73 | 88.13±0.33 | 92.61±0.28 | 95.81±0.11 |
| CHEBNET | 87.32±0.92 | 79.33±0.57 | 87.82±0.24 | 91.63±0.39 | 94.21±0.26 |
| GPR-GNN | 88.54±0.67 | 80.13±0.84 | 88.46±0.31 | 95.67±0.16 | 96.80±0.08 |
| BERNNET | 88.51±0.92 | 80.08±0.75 | 88.51±0.39 | 95.81±0.13 | 96.81±0.07 |
| CHEBNETII | 88.71±0.93 | 80.53±0.79 | 88.93±0.29 | 96.03±0.11 | 97.23±0.07 |
| SPECFORMER | 88.57±1.01 | 81.49±0.94 | 89.13±0.35 | 95.92±0.19 | 97.44±0.08 |
| EVENNET | 87.25±1.42 | 78.65±0.96 | 89.52±0.31 | 94.66±0.23 | 95.59±0.11 |
| FAVARD | 89.35±1.09 | 81.89±0.63 | 90.90±0.27 | 95.77±0.15 | 97.58±0.08 |
| PCNET | 90.02±0.62 | 81.76±0.78 | 91.30±0.38 | 96.33±0.15 | 97.62±0.08 |
| HALF-HOP | 88.73±1.22 | 80.33±0.66 | 89.86±0.36 | 95.13±0.21 | 95.75±0.13 |
| GCNII | 88.46±0.82 | 79.97±0.65 | 89.94±0.31 | 96.58±0.07 | 97.27±0.12 |
| TWIRLS | 88.57±0.91 | 80.07±0.94 | 88.87±0.43 | 95.43±0.04 | 97.17±0.07 |
| PDE-GCN | 88.62±1.03 | 79.98±0.97 | 89.92±0.38 | 95.35±0.19 | 96.89±0.08 |
| EGNN | 87.47±1.33 | 80.51±0.93 | 88.74±0.46 | 95.22±0.20 | 96.61±0.08 |
| TFE-GNN$_{con}$ | 90.11±1.27 | 82.39±0.96 | 90.94±0.29 | 93.55±1.56 | 97.62±0.23 |
| TFE-GNN$_{sum}$ | **90.73±1.11** | **82.83±1.24** | **91.66±0.51** | **96.96±0.17** | **98.85±0.13** |
| TFE-GNN\$TFE_1$ | 89.49±1.30 | 61.00±1.19 | 90.80±0.37 | 96.96±0.17 | 97.24±0.10 |
| TFE-GNN\$TFE_2$ | 90.15±1.75 | 82.83±1.24 | 91.66±0.51 | 96.57±0.16 | 97.19±0.19 |
| TFE-GNN$_{rw+sum}$ | 89.57±1.26 | 81.92±1.14 | 90.96±0.49 | 95.70±0.53 | 97.73±0.16 |

filters the graph signal $X$ in the following way:

$$
Z = \begin{cases}
\vartheta_1 Z_{lp} + \vartheta_2 Z_{hp} = (\vartheta_1 TFE_1 + \vartheta_2 TFE_2) \cdot X = (\vartheta_1 \sum_{i=0}^{K_{lp}} \omega_i (H_{lp})^i + \vartheta_2 \sum_{j=0}^{K_{hp}} \omega'_j (H_{hp})^j) \cdot X \\
\\
\vartheta_1 Z_{lp} \| \vartheta_2 Z_{hp} = (\vartheta_1 TFE_1 \| \vartheta_2 TFE_2) \cdot X = (\vartheta_1 \sum_{i=0}^{K_{lp}} \omega_i (H_{lp})^i \| \vartheta_2 \sum_{j=0}^{K_{hp}} \omega'_j (H_{hp})^j) \cdot X,
\end{cases}
\tag{15}
$$

where $\|$ denotes concatenation and has lower arithmetic priority than addition, subtraction, multiplication and division.

## 4.1 Supervised Node Classification

**Setting and baselines.** We compare TFE-GNN to a series of SOTA models for full-supervised node classification on datasets with random splits, including 11 polynomial approximation filter methods GCNs (Kipf & Welling, 2016), ARMA (Bianchi et al., 2021a), APPNP (Klicpera et al., 2018), ChebNet (Defferrard et al., 2016), GPR-GNN (Chien et al., 2021), BernNet (He et al., 2021a), ChebNetII (He et al., 2022), SPECFORMER (Bo et al., 2023a), EvenNet (Lei et al., 2022), FavardGNN/OptBasisGNN (Guo & Wei, 2023), and PCNet (Li et al., 2023). We also add 5 competitive SOTA models Half-Hop (Azabou et al., 2023), GCNII (Chen et al., 2020), TWIRLS (Yang et al., 2021), PDE-GCN (Eliasof et al., 2021), and EGNN (Zhou et al., 2021). We randomly split each class of nodes into 60%, 20%, and 20% as training, validation, and testing sets for full-supervised node classification and all models share the same ten random splits for a fair comparison.

For TFE-GNN, we set the hidden units to be 64 or 512 (Squirrel, Chaneleon, roman-empire, and amazon-rating), the number of early stoppings is 200 and the number of epochs is 1000 for all datasets. We employ the *ReLu* as an activation function for $f_{mlp}$. We use the officially released code for GCNII, GPR-GNN, BernNet, etc and use the Deep Graph Library implementations for other models, such as GCNs, APPNP, ChebNet, etc. More experimental details of hyper-parameters and code URLs are listed in Appendix C.

**Results.** Tables 2 and 3 report the results of the different models on all datasets and gives mean classification accuracy and standard deviation over ten random splits, where the bolded numbers indicate the best results, "Favard" denotes FavardGNN/OptBasisGNN, TFE-GNN$_{sum}$ denote that

Table 3: Mean accuracy of different models on datasets for full-supervised node classification.

| Datasets | Cora-Full | Chameleon | Squirrel | Wisconsin | Texas | Cornell |
|---|---|---|---|---|---|---|
| $ehr$ | 0.57 | 0.23 | 0.22 | 0.21 | 0.06 | 0.30 |
| MLP | 52.45±0.64 | 46.59±1.84 | 31.01±1.18 | 86.55±2.36 | 86.81±2.24 | 84.15±3.05 |
| GCNs | 66.04±0.38 | 60.81±2.95 | 45.87±0.88 | 74.19±3.15 | 76.97±3.97 | 65.78±4.16 |
| ARMA | 63.53±0.66 | 60.21±1.00 | 36.27±0.62 | 87.25±1.63 | 83.97±3.77 | 85.62±2.13 |
| APPNP | 59.85±0.54 | 52.15±1.79 | 35.71±0.78 | 91.08±1.79 | 90.64±1.70 | 91.52±1.81 |
| ChebNet | 58.65±0.74 | 59.51±1.25 | 40.81±0.42 | 84.19±2.58 | 86.28±2.62 | 83.91±2.17 |
| GPR-GNN | 71.86±0.29 | 67.49±1.38 | 50.43±1.89 | 91.71±1.62 | 92.91±1.32 | 91.57±1.96 |
| BernNet | 72.01±0.26 | 68.53±1.68 | 51.39±0.92 | 92.45±1.22 | 92.62±1.37 | 92.13±1.64 |
| ChebNetII | 72.11±0.24 | 71.37±1.01 | 57.72±0.59 | 93.72±1.27 | 93.28±1.47 | 92.30±1.48 |
| SPECFORMER | 71.84±0.26 | 74.72±0.19 | 64.64±0.81 | 92.98±1.84 | 92.77±2.37 | 91.86±2.69 |
| EvenNet | 70.04±0.47 | 67.57±1.52 | 50.36±0.93 | 93.55±1.68 | 93.77±1.73 | 92.13±1.71 |
| Favard | 72.39±0.34 | 74.26±0.74 | 63.62±0.76 | 93.33±1.95 | 91.87±3.11 | 92.06±2.96 |
| PCNet | 72.35±0.26 | 73.55±1.26 | 63.53±0.26 | 94.26±1.85 | 92.78±1.80 | 93.83±1.91 |
| Half-Hop | 72.55±0.31 | 62.98±3.35 | 45.25±1.52 | 87.59±1.77 | 85.95±6.42 | 74.60±6.06 |
| GCNII | 66.70±0.85 | 63.44±0.85 | 41.96±1.02 | 85.66±1.95 | 80.46±5.91 | 84.26±2.13 |
| TWIRLS | 68.88±0.22 | 50.21±2.97 | 39.63±1.02 | 91.53±2.81 | 91.31±3.36 | 89.83±2.29 |
| PDE-GCN | 71.37±0.35 | 66.01±1.56 | 48.73±1.06 | 92.85±1.67 | 93.24±2.03 | 89.73±1.35 |
| EGNN | 71.51±0.27 | 51.55±1.73 | 35.81±0.91 | 83.76±1.64 | 81.34±1.56 | 82.09±1.16 |
| TFE-GNN$_{con}$ | **74.12±0.40** | 77.03±1.47 | 71.47±1.15 | 96.00±1.73 | 93.11±4.26 | **94.26±2.77** |
| TFE-GNN$_{sum}$ | 73.60±0.21 | **77.16±1.41** | **72.27±1.32** | **97.38±1.42** | **94.87±2.66** | 93.11±2.96 |
| TFE-GNN\$TFE_1$ | 64.09±0.41 | 76.63±2.20 | 57.06±1.03 | 92.97±1.38 | 93.44±1.80 | 93.11±2.96 |
| TFE-GNN\$TFE_2$ | 73.60±0.21 | 61.05±2.45 | 41.91±0.69 | 81.62±8.40 | 68.36±7.10 | 93.11±2.96 |
| TFE-GNN$_{rw+sum}$ | 72.81±0.51 | 69.26±4.84 | 56.93±1.04 | 96.00±1.34 | 93.28±2.69 | 90.98±3.38 |

$EM_3$ of TFE-GNN is summation, and TFE-GNN$_{con}$ denote that $EM_3$ of TFE-GNN is concatenation. The experimental results are taken from ChebNetII, Half-Hop, SPECFORMER, EvenNet, and avardGNN/OptBasisGNN, when they report relevant results, and the remaining results are reproduced by us. Tables 2 and 3 illustrate that ChebNet starts to outperform GCNs when there is more training data available, which suggests the validity of the Chebyshev approximation. TFE-GNN achieves new state-of-the-art results on all datasets. TFE-GNN outperforms some SOTA models with similar results on Cora and Citeseer, including GPR-GNN (Chien et al., 2021), BernNet (He et al., 2021a), ChebNetII (He et al., 2022) and Half-Hop (Azabou et al., 2023). Notably, TFE-GNN outperforms ChebNetII on Chameleon and Squirrel by 5.79% and 14.85%, respectively, which amounts to performance improvements of 8% and 26%. TFE-GNN achieves exciting results on Physics (strong homophily) and Wisconsin (strong heterophily) for full-supervised node classification, already close to 100% accuracy. We put the relevant settings and results of TFE-GNN for semi-supervised node classification in Appendix B.2, due to space limitations.

## 4.2 Ablation Study

We conduct experiments to investigate the (joint) contributions of TFE-GNN's components. The last three rows of Tables 2 and 3 report the results of the ablation experiments. The symbol "TFE-GNN\$TFE_1$" means $\vartheta_1 = 0$ or $K_{lp} = 0$, "TFE-GNN\$TFE_2$" means $\vartheta_2 = 0$ or $K_{hp} = 0$, and "TFE-GNN$_{rw+sum}$" indicates that TFE-GNN$_{sum}$ uses the random walk normalized Laplacian $L_{rw} = D^{-1}L = I - D^{-1}A$ as the high-pass graph filter $H_{hp}$. Partial ablation experiments yielded the same results due to the choice of hyperparameters $K_{lp}$ and $K_{hp}$. We observe that TFE-GNN\$TFE_1$ performs worse under strong homophily, while TFE-GNN\$TFE_2$ performs worse under strong heterophily. TFE-GNN with all graph filters achieves the best results, which suggests that it can match various graph structure.

## 4.3 Generalization Analysis

We verify the generalization of TFE-GNN by analyzing its cross-entropy loss in the training and validation sets on Cora, and a smaller gap between the two losses indicates a better generalization of the model (Feng et al., 2020). Figure 2 shows the significant gap between the training and validation losses for ChebNet, ChebNetII, and PCNet, which indicates a possible overfitting and diminishes their generalization. The validation loss of TFE-GNN is much closer to its training loss and the early stopping mechanism allows the model to carry less stable losses. More generalization analyses are reported in Appendix B.4.

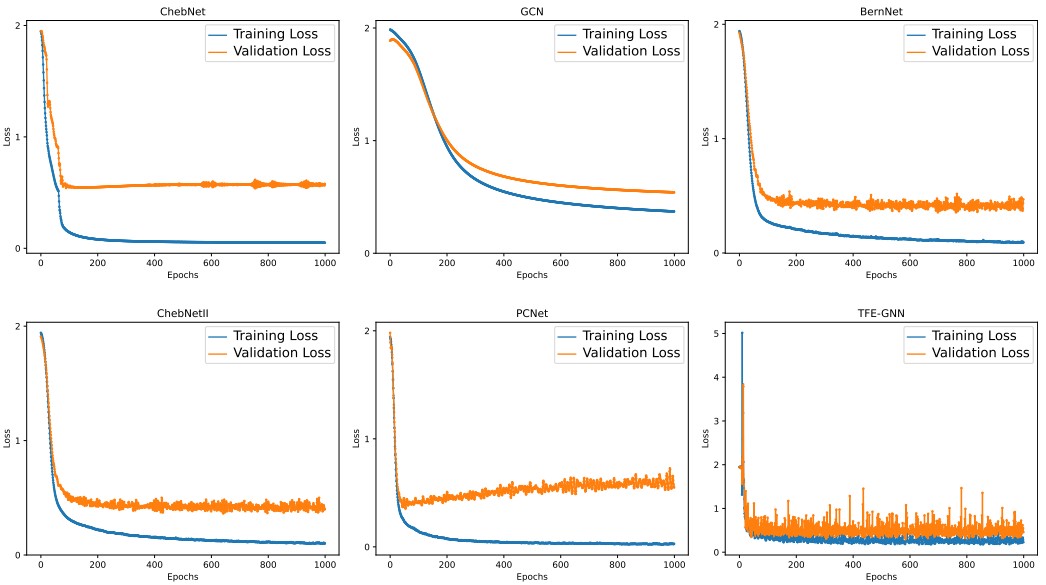

Figure 2: Generalization on Cora.

## 4.4 Loss Oscillation Analysis

We try to explain the reason for the oscillation in TFE-GNN's losses in Figure 2. The learning rate controls the step size which in turn affects the loss optimization. The large learning rate (= 0.1) is responsible for the oscillations of the validation loss in Figure 2. Figure 3 shows that the loss is stable when the learning rate is 0.001. The early stopping mechanism allows TFE-GNN to carry less stable losses and losses do not fall into unacceptable local minimum.

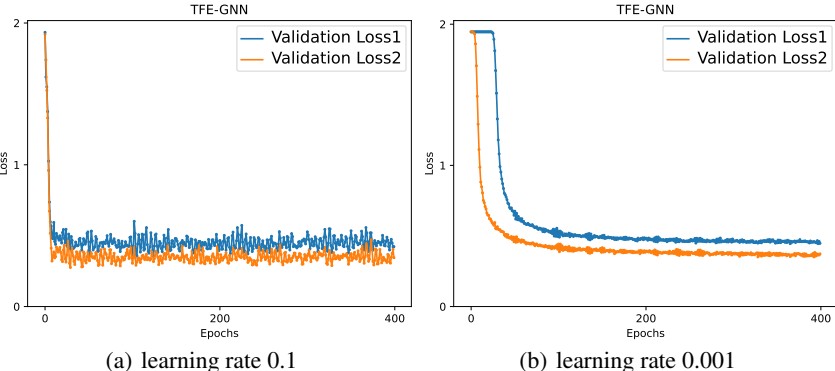

(a) learning rate 0.1        (b) learning rate 0.001

Figure 3: Verification loss at different learning rates, keeping the rest of the parameters constant. There are two validation loss curves on each subfigure, and each loss is the average of five experiments.

## 5 Conclusions

We propose TFE-GNN with triple filter ensembles (TFE) to solve three progressive problems. TFE-GNN extracts homophily and heterophily from graphs with different homophily levels adaptively while utilizing the initial features, which motivates it to match various graph structure. We theoretically prove that TFE-GNN can learn arbitrary filters and is a combination of two polynomial-based spectral GNNs. Experiments show that TFE-GNN achieves new state-of-the-art performance on various real-world datasets. In the future, we will dig deeper into ensemble methods of triple filter ensembles and expect to further improve the performance of TFE-GNN.

## Acknowledgments and Disclosure of Funding

We would like to thank the School of Computer Science of Guangzhou University and the School of Telecommunication of Tongji University for their help, including the experimental environment, office location, and writing guidance. This work was supported in part by the Education Bureau of Guangzhou Municipality under Grant 2024312243 and in part by the National Natural Science Foundation of China under Grant 62202507.

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

# A   Proofs and Limitations

## A.1   Proofs of Theorems 1 and 2

**Theorem 1.**  *TFE-GNN and ChebNet can be transformed into each other under the following conditions, (1) learning the proper coefficients $\omega$, $\omega'$, $\vartheta$ and ChebNet' coefficient $\theta$, (2) the ensemble methods $EM_1$, $EM_2$ and $EM_3$ take **summation**, the base high-pass filter $H_{hp}$ takes the symmetric normalized Laplacian $L_{sym}$ and the base low-pass filter $H_{lp}$ takes the affinity (transition) matrix of $L_{sym}$, and (3) $K_{hp} = K_{lp} = K - 1$. Thus, TFE-GNN can also learn arbitrary filters.*

*proof:* We first prove that ChebNet can be transformed into TFE-GNN under conditions (1), (2), and (3). Instead of ChebNet, we use its excellent version Chebyshev basis (Equation 7). We change Equation 7 to the following form that does not affect approximation ability:

$$\hat{Y} = softmax(f_{mlp}(\sum_{k=0}^{K} \theta_k T_k(\tilde{L})X)). \tag{16}$$

Equation 16 can be rewritten in the form of equal approximation ability using the learnable parameters (constants) $\vartheta_1$ and $\vartheta_2$:

$$\hat{Y} = softmax(f_{mlp}(\vartheta_1 \sum_{k=0}^{K} \theta_k T_k(\tilde{L})X + \vartheta_2 \sum_{k=0}^{K} \theta_k T_k(\tilde{L})X)) = softmax(f_{mlp}(\vartheta_1 \tilde{Y} + \vartheta_2 \tilde{Y})). \tag{17}$$

We parse and reconstruct the key parts of Equation 17 by unfolding and integrating technologies, i.e., analyzing $\sum_{k=0}^{K} \theta_k T_k(\tilde{L})X = \tilde{Y}$. We get the follow equation by unfolding the summation and Chebyshev polynomial $T(k)(\tilde{L})$:

$$\begin{aligned}
\tilde{Y} &= (\theta_0 T_0(\tilde{L}) + \theta_1 T_1(\tilde{L}) + \cdots + \theta_{K-1} T_{K-1}(\tilde{L}))X \\
&= (\theta_0 I + \theta_1 \tilde{L} + \theta_2(2\tilde{L} - I) + \cdots + \theta_{K-1}(2\tilde{L}(T_{K-2}(\tilde{L})) - T_{K-3}(\tilde{L})))X,
\end{aligned} \tag{18}$$

where $T_0(\tilde{L}) = I$, $T_1(\tilde{L}) = \tilde{L}$, $T_2(\tilde{L}) = 2\tilde{L}^2 - I$, and so on. Equation 18 can be viewed as a $(K-1)$-th order polynomial of $\tilde{Y}$ with respect to $\tilde{L}$. We then integrate the coefficients of different orders $\tilde{L}$ in Equation 18:

$$\begin{aligned}
\tilde{Y} &= ((\theta_0 - \theta_2 + \theta_4 - \theta_6 + \cdots)I + \cdots + \theta_{K-1} 2^{K-2} \tilde{L}^{K-1})X \\
&= (\theta'_0 I + \theta'_1 \tilde{L} + \cdots + \theta'_{K-1} \tilde{L}^{K-1})X,
\end{aligned} \tag{19}$$

where the first coefficient of the polynomial $\theta'_0 = (\theta_0 - \theta_2 + \theta_4 - \theta_6 + \cdots)$ in Equation 19, the coefficient of the highest order $\theta'_{K-1} = \theta_{K-1} 2^{K-2}$, and so on. We can calculate the relationship between coefficients $\theta'_k$ and $\theta_k (0 < k < K - 1)$ based on the generalized form of the Chebyshev polynomials.

$$T_K(\boldsymbol{x}) = \sum_{k=0}^{\lfloor n/2 \rfloor} [(-1)^k \sum_{j=k}^{\lfloor n/2 \rfloor} \binom{n}{2j}\binom{j}{k}]\boldsymbol{x}^{K-2k}, \tag{20}$$

where $\lfloor n/2 \rfloor$ denotes downward rounding, and $\binom{n}{2j}$ denotes combinatorial numbers. We get the connection between $\hat{Y}_1$ and $Z_{hp}$ according to the given conditions (1), (2), and (3):

$$\tilde{Y} = (\omega'_0 I + \omega'_1 (H_{hp})^1 + \cdots + \omega'_{K_{hp}}(H_{hp})^{K_{hp}})X = TFE_2 \cdot X = Z_{hp}, \tag{21}$$

where the conversion of the coefficients $\theta'$ to $\omega'$ are realized through equation $\tilde{L} = 2L_{sym}/\lambda_{max} - I$. We continue to unfold Equation 19:

$$\begin{aligned}
\tilde{Y} &= (\theta'_0 I + \theta'_1 (\frac{2L}{\lambda_{max}} - I) + \cdots + \theta'_{K-1}(\frac{2L}{\lambda_{max}} - I)^{K-1})X \\
&= (\theta'_0 I + \theta'_1 (\frac{2 - \lambda_{max}}{\lambda_{max}}I - \frac{2}{\lambda_{max}}(D^{-\frac{1}{2}}AD^{-\frac{1}{2}})) + \cdots + \\
&\quad \theta'_{K-1}(\frac{2 - \lambda_{max}}{\lambda_{max}}I - \frac{2}{\lambda_{max}}(D^{-\frac{1}{2}}AD^{-\frac{1}{2}}))^{K-1})X \\
&= (\theta''_0 I + \theta''_1 (D^{-\frac{1}{2}}AD^{-\frac{1}{2}}) + \cdots + \theta''_{K-1}(D^{-\frac{1}{2}}AD^{-\frac{1}{2}})^{K-1})X,
\end{aligned} \tag{22}$$

where $\theta_0'' = \theta_0' + \frac{2-\lambda_{max}}{\lambda_{max}}\theta_1' + \cdots + (\frac{2-\lambda_{max}}{\lambda_{max}})^{K-1}\theta_{K-1}'$, and $\theta_{K-1}'' = (\frac{2}{\lambda_{max}})^{K-1}\theta_{K-1}'$. We get the connection between $\tilde{Y}$ and $Z_{lp}$ according to the given conditions (1), (2), and (3):

$$
\begin{aligned}
\tilde{Y} &= (\theta_0'' I + \theta_1''(H_{lp}) + \cdots + \theta_{K-1}''(H_{lp})X \\
&= (\omega_0 I + \omega_1(H_{lp})^1 + \cdots + \omega_{K_{lp}}(H_{hp})^{K_{hp}})X = TFE_1 \cdot X = Z_{lp},
\end{aligned}
\tag{23}
$$

where $\omega = \theta''$. We get a new conclusion about $\hat{Y}$:

$$
\hat{Y} = softmax(f_{mlp}(\vartheta_1 Z_{lp} + \vartheta_2 Z_{hp})) = softmax(f_{mlp}(Z)) = softmax(\tilde{Z}) = Z^*. \tag{24}
$$

ChebNet and its excellent version Chebyshev basis are essentially the same (Defferrard et al., 2016), except that Chebyshev basis implements the decoupling (He et al., 2022). Therefore, we proved that ChebNet can be transformed into TFE-GNN under conditions (1), (2), and (3), and vice versa.

**Theorem 2.** *TFE-GNN can be rewritten in the following form, with certain conditions to be satisfied, which is a combination of two polynomial-based learnable spectral GNNs: $Z^* = softmax(f_{mlp}(\vartheta_1 \sum_{k=0}^{K'} \bar{\theta}_k^1 P_k^1(\bar{H}_{gf}^1)^k X \bigoplus \vartheta_2 \sum_{k=0}^{K''} \bar{\theta}_k^2 P_k^2(\bar{H}_{gf}^2)^k X))$, where $P_k$ denote polynomials used for approximation, $\bar{\theta}$ are the learnable coefficients, $\bar{H}_{gf}$ denote graph filters, and $\bigoplus$ denotes $EM_3$. Conditions are (1) learning the proper coefficients $\omega$, $\omega'$, $\vartheta$, $\bar{\theta}^1$, and $\bar{\theta}^2$, (2) the ensemble methods $EM_1$, $EM_2$ take* **summation** *and $EM_3$ takes ensemble method capable of preserving the properties of the model, such as* **summation** *and* **concatenation**, *the base high-pass filter $H_{hp}$ takes $\bar{H}_{gf}^2$ and the base low-pass filter $H_{lp}$ takes $\bar{H}_{gf}^1$, and (3) $K_{lp} = K'$ and $K_{hp} = K''$. Thus, TFE-GNN can match various graph structures adaptively.*

*proof*: The proof of Theorem 2 is similar to that of Theorem 1. First, we unfold the polynomials $P_k^1$ and $P_k^2$ in $Z^*$ to get a combination of $K'$-th order polynomial with respect to $\bar{H}_{gf}^1$ and $K''$-th order polynomial with respect to $\bar{H}_{gf}^2$, which is similar to Equation 18. $P_k$ can be the Chebyshev, Bernstein, even, Possion-Charlier polynomials in ChebNet (Defferrard et al., 2016), BernNet (He et al., 2021a), EvenNet (Lei et al., 2022), PCNet (Li et al., 2023), and so on. We then sum their coefficients using $(\bar{H}_{gf}^1)^k(k \in [0, K'])$ and $(\bar{H}_{gf}^2)^k(k \in [0, K''])$ as variables respectively, which is similar to Equations 19 and 22. Finally, we replace $\bar{\theta}^1$ and $\bar{\theta}^2$ with $\omega$ and $\omega'$, which is similar to Equations 21 and 23. Therefore, TFE-GNN is a combination of two polynomial-based learnable spectral GNNs. Theorems 1 and 2 state that TFE-GNN can learn any filter and matches various graph structure and extracts homophily and heterophily from graphs adaptively.

Table 4: Mean classification accuracy of models at different $H_{lp}$ and $H_{hp}$.

| DATASETS | $(H_{lp}, H_{hp})$ | | | |
|---|---|---|---|---|
| | (5, 0) | (0, 5) | (5, 5) | (10, 10) |
| CITESEER | 82.36±1.35 | 77.89±1.03 | 82.32±1.00 | 81.68±1.41 |
| WISCONSIN | 96.38±2.20 | 94.88±2.05 | 95.75±2.81 | 96.00±2.49 |

### A.2 Limitations

We design triple filter ensembles to generate a TFE-Conv, which can match various graph structures and extract the initial information, homophily and heterophily adaptively. The learnable coefficients $\vartheta_1$ and $\vartheta_2$ and customized hyperparameters $K_{lp}$ and $K_{hp}$ enhance the performance of TFE-GNN while creating some limitations. In particular, $K_{lp}$ and $K_{hp}$ rely on one's experience, and it is solved by selecting the appropriate $K_{lp}$ and $K_{hp}$ according to the homophily level of datasets. For example, we choose a larger $K_{lp}$ for datasets with strong homophily while choosing a larger $K_{hp}$ for datasets with strong heterophily. However, the real-world dataset cannot determine its homophily level due to the presence of unlabeled nodes. We are able to determine its approximate homophily level based on training set labels or dataset knowledge, which may be in error.

It is conceivable that there are other limitations of the TFE-GNN, such as the choice of the base filter, which affect the performance of the model. In fact, none of $K_{lp}$, $K_{hp}$, $H_{lp}$ and $H_{hp}$ is fatally weakening the performance of TFE-GNN, as long as we make a suitable choice based on the general phenomenon of polynomial-based spectral GNNs. For example, $H_{hp}$ taking the symmetric normalized Laplacian $L_{sym}$ or the random walk normalized Laplacian $L_{rw}$, $H_{hp}$ taking their affinity

(transition) matrix $L_{sym}^a$ or $L_{rw}^a$, $K_{lp}$ and $K_{hp}$ taking the interval [1,10]. Table 4 show mean classification accuracy of TFE-GNN ($H_{hp}$ taking $L_{sym}$) at different $H_{lp}$ and $H_{hp}$, where $n_1$ and $n_2$ in $(n_1, n_2)$ are the values of $H_{lp}$ and $H_{hp}$, respectively.

Table 5: Mean classification accuracy of different models for semi-supervised node classification.

| DATASETS | CORA | CITESEER | PUBMED | WISCONSIN | TEXAS | CORNELL |
|---|---|---|---|---|---|---|
| MLP | 58.88±0.62 | 56.97±0.54 | 73.15±0.28 | 47.09±4.25 | 45.03±2.45 | 46.18±5.10 |
| GCNs | 81.32±0.18 | 71.77±0.21 | 79.15±0.18 | 35.34±3.72 | 32.42±2.23 | 35.57±3.55 |
| ARMA | 83.15±0.54 | 71.31±0.36 | 78.75±0.14 | 45.81±3.46 | 47.84±3.35 | 30.89±4.23 |
| APPNP | 83.52±0.24 | 72.09±0.25 | 80.23±0.15 | 47.31±3.05 | 46.31±3.01 | 45.73±4.85 |
| CHEBNET | 80.54±0.38 | 70.35±0.33 | 75.52±0.75 | 28.62±4.28 | 28.55±3.28 | 25.54±3.42 |
| GPR-GNN | 83.95±0.22 | 70.92±0.57 | 78.97±0.27 | 45.91±4.12 | 45.76±3.78 | 43.42±4.95 |
| BERNNET | 82.15±0.32 | 72.24±0.25 | 79.65±0.25 | 49.93±3.15 | 48.31±3.17 | 46.64±5.62 |
| CHEBNETII | 83.67±0.33 | 72.75±0.16 | 80.48±0.23 | 53.16±3.17 | 54.68±3.87 | 50.92±5.49 |
| EVENNET | 82.85±0.32 | 72.57±0.42 | 79.36±0.31 | 54.55±2.68 | 53.77±3.73 | 50.64±5.71 |
| FAVARD | 83.61±0.30 | 73.16±0.24 | 80.52±0.26 | 54.13±2.95 | 55.23±3.41 | 51.43±4.96 |
| TFE-GNN$_{con}$ | 83.62±0.39 | 72.77±0.64 | 80.41±0.49 | **63.12±4.91** | 64.34±3.03 | 57.92±2.32 |
| TFE-GNN$_{sum}$ | **84.11±0.20** | 74.53±0.25 | **80.78±0.21** | 58.71±2.97 | **67.24±1.19** | **58.21±4.27** |
| TFE-GNN$_{rw+sum}$ | 83.13±0.62 | **74.91±0.22** | 79.73±0.48 | 43.33±3.25 | 61.27±2.57 | 55.66±4.22 |

# B    More Experimental Results

## B.1    Additional experiments

**Datasets and Setting.** We illustrate the generalization and scalability of TFE-GNN with more intuitive experiments. We select four new datasets (Platonov et al., 2023; Lim et al., 2021a) and conduct relevant experiments to further explain why TFE-GNN has better generalization, including roman-empire, amazon-rating, and two large graphs (Lim et al., 2021a) fb100-Penn94 and genius, whose edge homophily are 0.05, 0.38, 0.47, and 0.62, respectively. Experimental results show that TFE-GNN achieves competitive performance on these datasets with results of 75.87%, 52.21%, 84.76% and 89.32%, respectively. TFE-GNN achieves competitive rankings on all datasets, with the best performance on fb100-Penn94, outperforming most spectral GNNs on roman-empire, and topping the rest of the datasets. These additional experimental results further validate the generalization ability of TFE-GNN on both homophilic and heterophilic datasets: TFE-GNN can generalize well on graphs with different edge homophily levels. Dataset statistics are reported in Table 6 and the hyperparameters are reported in Table 7. Note that we use the same dataset splits as in the article (Platonov et al., 2023; Lim et al., 2021a).

## B.2    Semi-supervised node classification

**Setting and baselines.** We compare TFE-GNN to a series of SOTA models for semi-supervised node classification on real-world datasets, including GCNs (Kipf & Welling, 2016), ARMA (Bianchi et al., 2021a), APPNP (Klicpera et al., 2018), ChebNet (Defferrard et al., 2016), GPR-GNN (Chien et al., 2021), BernNet (He et al., 2021a), ChebNetII (He et al., 2022), EvenNet (Lei et al., 2022), and FavardGNN/OptBasisGNN (Guo & Wei, 2023). We employ the standard and popular training/validation/testing split method (Kipf & Welling, 2016; Klicpera et al., 2018) on three citation networks for semi-supervised node classification, i.e., Cora, Citeseer, and Pubemd, with 20 nodes per class for training, 500 and 1,000 nodes for validation and testing. We split each class of nodes into 2.5%, 2.5%, and 95% as training, validation, and testing sets for other datasets and all models share the same ten random splits for a fair comparison. The seed used for dataset splitting is 42 or 1941488137 (He et al., 2022).

**Results.** Table 5 reports the results of the different models on all datasets for semi-supervised node classification and gives mean classification accuracy and standard deviation over ten foxed splits. Table 5 shows that TFE-GNN achieves state-of-the-art performance on 6 datasets, where TFE-GNN$_{con}$ and TFE-GNN$_{rw+sum}$ also show strong competitiveness, which is similar to the experimental results of full-supervised node classification. TFE-GNN outperforms FavardGNN/OptBasisGNN (Guo & Wei, 2023) on Wisconsin, Texas, and Corenell (strong heterophily) by 8.99%, 12.01%, and 6.78%, respectively, which amounts to performance improvements of 17%, 22%, and 13%.

Table 6: Additional dataset statistics.

| DATASETS | ROMAN-EMPIRE | AMAZON-RATING | FB100-PENN94 | GENIUS |
|---|---|---|---|---|
| NODES | 22662 | 24492 | 41554 | 421961 |
| EDGES | 32927 | 93050 | 1362229 | 984979 |
| FEATURES | 300 | 300 | 5 | 12 |
| CLASSES | 18 | 5 | 2 | 2 |
| $ehr$ | 0.05 | 0.38 | 0.47 | 0.62 |

Table 7: The hyper-parameters of TFE-GNN for additional datasets.

| DATASETS | ROMAN-EMPIRE | AMAZON-RATING | FB100-PENN94 | GENIUS |
|---|---|---|---|---|
| $optim$ | ADAM | RMSPROP | ADAMW | ADAMW |
| $K_{lp}$ | 4 | 0 | 7 | 8 |
| $K_{hp}$ | 1 | 9 | 0 | 6 |
| $drop_{pro}$ | 0.0 | 0.4 | 0.9 | 0.0 |
| $drop_{lin}$ | 0.3 | 0.8 | 0.5 | 0.9 |
| $\eta$ | 0.4 | 0.5 | 0.5 | 0.4 |
| $lr_{ada}$ | 0.5 | 0.5 | 0.01 | 0.05 |
| $wd_{ada}$ | 0.05 | 0.0005 | 0.0005 | 0.0 |
| $lr_{adae}$ | 0.05 | 0.05 | 0.01 | 0.005 |
| $wd_{adae}$ | 0.1 | 0.05 | 0.0 | 0.0 |
| $lr_{lin}$ | 0.05 | 0.005 | 0.01 | 0.005 |
| $wd_{lin}$ | 0.0005 | 0.005 | 0.1 | 0.05 |

### B.3 Visualization Analysis

The t-SNE visualization in Figure 4 demonstrates that the graph convolution $TFE_1$(b) can extract meaningful patterns on Citeseer (strong homophily), which the graph convolution $TFE_2$(c) is not able to capture. The t-SNE visualization in Figure 5 demonstrates that $TFE_2$ is better than $TFE_1$ at extracting the information used for categorization on Squirrel (strong heterophily). The output of TFE-GNN(d) shows clearer boundaries among classes (colors) than that of $TFE_1$ and $TFE_2$. These visualization findings validate the competitiveness of TFE-GNN and echo its leading performance on node classification.

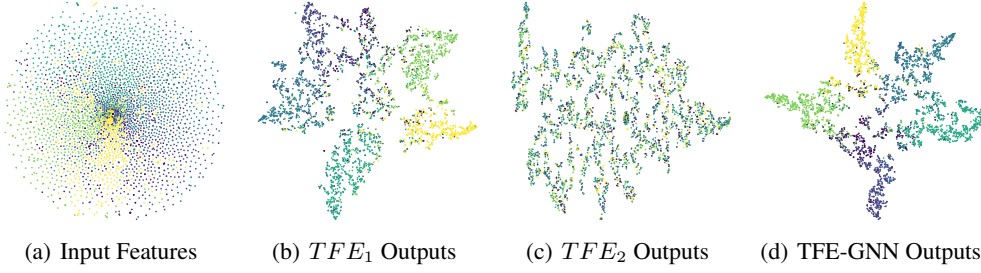

(a) Input Features     (b) $TFE_1$ Outputs     (c) $TFE_2$ Outputs     (d) TFE-GNN Outputs

Figure 4: Visualization of different models on Citeseer, which uses a dimensionality reduction method t-SNE with 1000 iterations.

### B.4 Complementary Generalization Analysis

We add some experiments to analyze TFE-GNN's generalization. These newly designed generalization experiments use the early stopping mechanism, i.e., no longer train for 1000 epochs as fixed as in Figure 2. Figure 6 reports the results for ChebNet(a), GCN(b), BernNet(c), ChebNetII(d), PCNet(e) and TFE-GNN(f), and a smaller gap between the two losses indicates a better generalization of the model. As in Figure 2, the gap between the training and validation losses for GCN(b) is less than ChebNet(a) and ChebNetII(c) and larger than TFE-GNN(f). We can observe that the validation loss

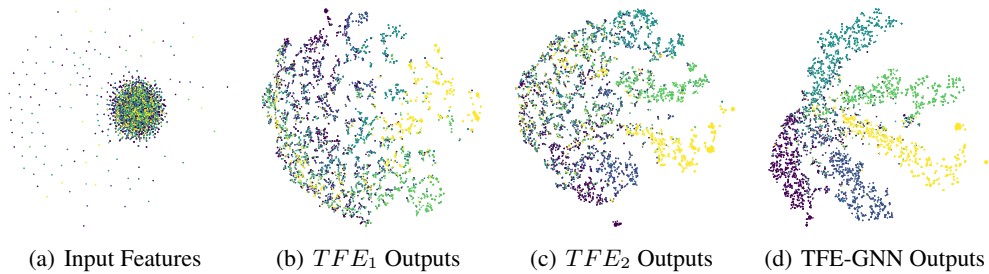

(a) Input Features      (b) $TFE_1$ Outputs      (c) $TFE_2$ Outputs      (d) TFE-GNN Outputs

Figure 5: Visualization of different models on Squirrel, which uses a dimensionality reduction method t-SNE with 1000 iterations.

of TFE-GNN(f) is much closer to its training loss in the six subplots of Figure 6, although its losses are more volatile. This observation demonstrates that TFE-GNN's generalization is best when it achieves state-of-the-art classification performance because the early stopping mechanism allows TFE-GNN to carry less stable losses.

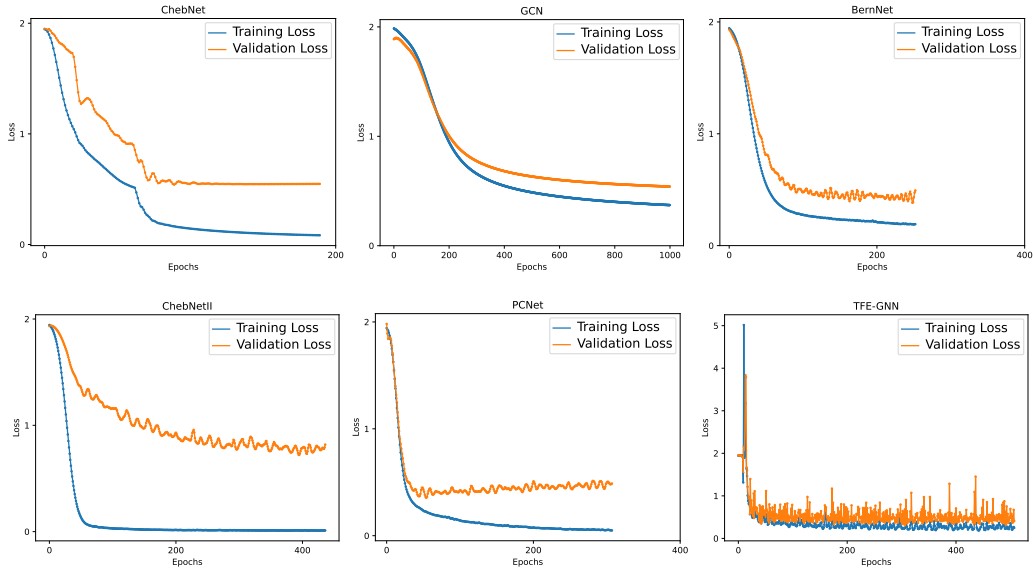

Figure 6: Generalization of models on Cora.

## B.5 Hyper-parameters Analysis

TFE-GNN yields two hyper-parameters $K_{lp}$ and $K_{hp}$ in addition to those associated with the neural network, since we parameterize $\omega$, $\omega'$, and $\vartheta$ as learnable coefficients. We conduct 200 experiments to obtain the interactions between $K_{lp}$ and $K_{hp}$ and the objective values corresponding to model accuracy. We draw their contour plots on Cora and Chameleon. Figure 7 shows parametric contour, where the horizontal coordinate "hop1" denotes the hyper-parameter $K_{lp}$ and the vertical coordinate "hop2" denotes the hyper-parameter $K_{hp}$. The parametric contour in Figure 7 demonstrates that there is a difference in the distributions of the objective values (model accuracy) on Cora(a) and Chameleon(b), i.e., there is a large difference in the influence of "hop1" and "hop2" on homophily and heterophily graphs.

## B.6 Time Efficiency Analysis

We conduct time efficiency experiments to count training times for ChebNet (Defferrard et al., 2016), BernNet (He et al., 2021a), ChebNetII (He et al., 2022) and TFE-GNN. All experiments are carried out on the machine with Linux system, two NVIDIA Tesla V100 and twelve Intel(R) Xeon(R) Gold

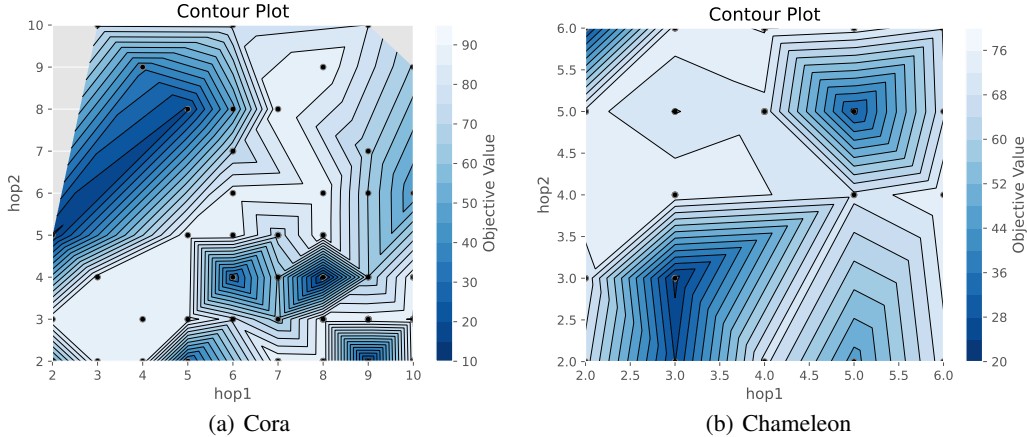

(a) Cora          (b) Chameleon

Figure 7: Parametric contours of Cora and Chameleon.

5220 CPU @2.20GHz. Table 8 shows the results of time efficient experiments on Cora, where TFE-GNN_zero indicates that $K_{lp}$ or $K_{hp}$ has and only a value of 0, TFE-GNN_ten indicates that hyper-parameter $K_{lp}$ or $K_{hp}$ has and only a value of 10, the other hyper-parameter has the value $K$ and TFE-GNN_all indicates that $K_{lp}$ and $K_{hp}$ have the same value $K$. We take the average of the times of ten training (100 epochs each) as time overheads (unit: second).

The columns in Table 8 imply the change in model-invariant training time as $K$ increases, and the rows imply the change in $K$-invariant training time as the model varies. Table 8 shows the variation in training time and the comparison between the different models. We observe that the training time of TFE-GNN is significantly smaller than that of ChebNetII when the value of $K$ is relatively large. Although BernNet and ChebNetII start with a low training time, they quickly catch up and surpass the other models as $K$ increases. Therefore, TFE-GNN has a clear advantage when a larger receptive field ($K$) is required.

Table 8: Time overheads (s) on Cora.

|  | CHEBNET | BERNNET | CHEBNETII | TFE-GNN_ZERO | TFE-GNN_TEN | TFE-GNN_ALL |
|---|---|---|---|---|---|---|
| $K$=10 | 10.88 | 1.89 | 3.27 | 3.03 | 5.60 | 5.60 |
| $K$=20 | 13.34 | 5.16 | 9.75 | 5.67 | 8.12 | 10.98 |
| $K$=40 | 22.30 | 17.19 | 35.77 | 10.85 | 13.44 | 21.82 |
| $K$=60 | 35.94 | 36.48 | 79.52 | 16.02 | 18.57 | 31.62 |
| $K$=80 | 56.38 | 64.28 | 137.60 | 21.07 | 23.89 | 42.00 |
| $K$=100 | 80.74 | 98.76 | 211.36 | 26.32 | 29.16 | 52.27 |

### B.7 TFE-GNN Versus Other GNNs with Similar Paradigms

TFE-GNN is more suitable for full-supervised node classification than semi-supervised node classification. The key technical differences: TFE-GNN is free from polynomial computation, coefficient constraints, and specific scenarios, compared to polynomial-based spectral GNNs and heterophily-specific Models (Platonov et al., 2023), such as ChebNetII (He et al., 2022) and PCNet (Li et al., 2023). We also describe the differences between TFE-GNN and several other methods (Li et al., 2024; Huang et al., 2024b,a).

PEGFAN (Li et al., 2024) sets up three optional feature matrices for the homophilic and heterophilic graphs, respectively, and performs a concatenation operation for the selected feature matrices. PEGFAN performs row normalization on the feature matrices that message passing and dimensionality-reducing outputs, and adds a new linear layer before $softmax$ after the activation function $ReLU$. These components increase network complexity of PEGFAN, whereas TFE-GNN is more concise and requires less space.

UniFilter (Huang et al., 2024b) establishes a link between the estimated graph homophily rate $\hat{h}$ and the propagation matrix $P$ by $\theta = \pi/2(1 - \hat{h})$, and forms heterophily bases $u_0, u_1, \ldots, u_K$ and

the rotation matrix $P_\theta$, in which learn coefficient $w$ discloses the significance of each frequency component in graph. The performance of UniFilter is affected by the $\hat{h}$ estimated from the training data labels, which seems to require that the training data label distribution is similar to the graph true global label distribution. The difference between TFE-GNN and UniFilter is that TFE-GNN rarely relies on such priori knowledge and conditions, which ensures its better adaptability and classification performance.

Flow2GNN (Huang et al., 2024a) is an interesting and novel attempt, which decomposes the original adjacency matrix into two matrices via a random binary matrix $G^{(l)}$ with elements that follow the Bernoulli distribution. Information flows within the nodes in two respective disentangled graphs with reduced heterophily, and then adaptively aggregates them with the strength estimation vector $p_i^{(l)}$ of information flow. TFE-GNN combines the well-performing and well-tested graph filters, while Flow2GNN is highly dependent on $G^{(l)}$ and $p_i^{(l)}$.

## C   More Experimental Details

**Hyper-parameters.** We provide more experimental details for reproducing the experiments. Table 9 shows the hyper-parameters of TFE-GNN on datasets for full-supervised node classification. Table 10 shows the hyper-parameters of TFE-GNN on datasets for semi-supervised node classification. The symbol $optim$ denotes the optimizer, $K_{lp}$ denotes the order of the low-pass graph filter, $K_{hp}$ denotes the order of the high-pass graph filter, $drop_{pro}$ denotes the dropout rate of input features, $drop_{lin}$ denotes the dropout rate of intermediate features, $lr_{ada}$ denotes the learning rate of the learnable coefficients $\omega$ and $\omega'$, $wd_{ada}$ denotes the weight decay of $\omega$ and $\omega'$, $lr_{adae}$ denotes the learning rate of $\vartheta$, $wd_{adae}$ denotes the weight decay of $\vartheta$, $lr_{lin}$ denotes the learning rate of MLP $f_{mlp}$, and $wd_{lin}$ denotes the weight decay of $f_{mlp}$.

Note that different DGL and PyTorch versions can affect model performance, and that fine-tuning of parameters, including but not limited to hyperparameters and seeds, may be required in order to reproduce the effect in this paper. For datasets roman-empire, amazon-rating, fb100-Penn94 and genius, this paper uses DGL 0.5.2 and PyTorch 1.5.1, and for other datasets, this paper uses DGL 0.9.0 and PyTorch 1.12.1.

**Baseline implementations.** We use the officially released code for GCNII, TWIRLS, GPR-GNN, BernNet, ChebNetII, H$_2$GCN, Haf-Hop, ARMA, EGNN, PDE-GCN, SPECFORMER, EvenNet, FavardGNN/OptBasisGNN,and PCNet. And we use the Deep Graph Library implementations for other models, such as GCNs, APPNP, ChebNet, etc. We did not spend a lot of time tuning parameters for these models. The code URLs are as follows.

Table 9: The hyper-parameters of TFE-GNN for full-supervised node classification.

| DATASETS | CORA | CITESEER | PUBMED | CS | CORA-FULL | PHYSICS | CHAMELEON | SQUIRREL | WISCONSIN | TEXAS | CORNELL |
|---|---|---|---|---|---|---|---|---|---|---|---|
| $optim$ | RMSPROP | RMSPROP | ADAM | RMSPROP | RMSPROP | RMSPROP | ADAM | ADAM | RMSPROP | RMSPROP | RMSPROP |
| $K_{lp}$ | 10 | 6 | 6 | 0 | 6 | 6 | 6 | 6 | 9 | 1 | 0 |
| $K_{hp}$ | 2 | 0 | 0 | 3 | 0 | 1 | 8 | 5 | 7 | 1 | 0 |
| $drop_{pro}$ | 0.5 | 0.4 | 0.2 | 0.4 | 0.2 | 0.2 | 0.4 | 0.6 | 0 | 0.1 | 0.3 |
| $drop_{lin}$ | 0.3 | 0.5 | 0.0 | 0.5 | 0.5 | 0.5 | 0.3 | 0.0 | 0.7 | 0.4 | 0.3 |
| $\eta$ | 0.5 | 0.5 | 0.5 | 0.5 | 0.5 | 0.5 | 0.3 | 0.3 | 0.4 | 0.3 | 0.3 |
| $lr_{ada}$ | 0.001 | 0.01 | 0.01 | 0.01 | 0.01 | 0.01 | 0.01 | 0.1 | 0.05 | 0.005 | 0.005 |
| $wd_{ada}$ | 0.5 | 0.5 | 0.05 | 0.1 | 0.0 | 0.0 | 0.1 | 0.05 | 0.0005 | 0.01 | 0.05 |
| $lr_{adae}$ | 0.1 | 0.005 | 0.01 | 0.05 | 0.0 | 0.0 | 0.0 | 0.0 | 0.001 | 0.01 | 0.0 |
| $wd_{adae}$ | 0.05 | 0.1 | 0 | 0.05 | 0.0 | 0.0 | 0.0 | 0.0 | 0.5 | 0.0005 | 0.0 |
| $lr_{lin}$ | 0.1 | 0.05 | 0.1 | 0.01 | 0.001 | 0.001 | 0.001 | 0.005 | 0.05 | 0.01 | 0.01 |
| $wd_{lin}$ | 0.0005 | 0.0005 | 0.0005 | 0.0005 | 0.01 | 0.01 | 0.0005 | 0.0 | 0.0005 | 0.0005 | 0.005 |

**Deep Graph Library:** https://docs.dgl.ai/en/0.6.x/guide/index.html

**GCNII:** https://github.com/chennnM/GCNII

**GPR-GNN:** https://github.com/jianhao2016/GPRGNN

**BernNet:** https://github.com/ivam-he/BernNet

**ChebNetII:** https://github.com/ivam-he/ChebNetII

**H$_2$GCN:** https://github.com/GemsLab/H2GCN

Table 10: The hyper-parameters of TFE-GNN for semi-supervised node classifications.

| DATASETS | CORA | CITESEER | PUBMED | WISCONSIN | TEXAS | CORNELL |
|---|---|---|---|---|---|---|
| $optim$ | ADAM | ADAM | RMSPROP | ADAM | RMSPROP | RMSPROP |
| $K_{lp}$ | 9 | 9 | 10 | 9 | 0 | 1 |
| $K_{hp}$ | 2 | 1 | 9 | 0 | 1 | 1 |
| $drop_{pro}$ | 0.8 | 0.1 | 0.9 | 0.3 | 0.4 | 0.6 |
| $drop_{lin}$ | 0.5 | 0.4 | 0.9 | 0.7 | 0.1 | 0.5 |
| $\eta$ | 0.5 | 0.4 | 0.4 | 0.4 | 0.3 | 0.5 |
| $lr_{ada}$ | 0.01 | 0.01 | 0.05 | 0.05 | 0.005 | 0.001 |
| $wd_{ada}$ | 0.1 | 0.0005 | 0 | 0.05 | 0.1 | 0.0005 |
| $wd_{adae}$ | 0.01 | 0.0 | 0.1 | 0.01 | 0.005 | 0.001 |
| $lr_{adae}$ | 0.005 | 0.01 | 0.1 | 0.01 | 0 | 0.0005 |
| $lr_{lin}$ | 0.005 | 0.001 | 0.005 | 0.05 | 0.01 | 0.05 |
| $wd_{lin}$ | 0.01 | 0.1 | 0.01 | 0.01 | 0 | 0.005 |

**TWIRLS:** https://github.com/FFTYYY/TWIRLS

**Haf-Hop:** https://github.com/nerdslab/halfhop

**ARMA:** https://github.com/xnuohz/ARMA-dgl

**EGNN:** https://github.com/Kaixiong-Zhou/EGNN

**PDE-GCN:** https://openreview.net/forum?id=wWtk6GxJB2x

**EvenNet:** https://github.com/Leirunlin/EvenNet

**SPECFORMER:** https://github.com/bdy9527/Specformer

**PCNet:** https://github.com/uestclbh/PC-Conv

**FavardGNN/OptBasisGNN:** https://github.com/yuziGuo/FarOptBasis

