# OpenReview forum: "Unifying Homophily and Heterophily for Spectral Graph Neural Networks via Triple Filter Ensembles"
_NeurIPS.cc/2024/Conference — NeurIPS 2024 poster_

### Official Review · Reviewer_zzRJ · 2024-07-12

**Soundness:** 3
**Presentation:** 3
**Contribution:** 3
**Rating:** 6
**Confidence:** 5

**Summary:**

This paper proposes a novel spectral GNN architecture, the TFE-GNN, which integrates homophily and heterophily using a triple filter ensemble (TFE) mechanism. The model aims to overcome the limitations of existing polynomial-based learnable spectral GNNs by adaptively extracting features from graphs with varying levels of homophily. The TFE-GNN combines low-pass and high-pass filters through three ensembles, enhancing the generalization and performance of the model. Theoretical analysis and experiments on real-world datasets demonstrate the effectiveness and state-of-the-art performance of TFE-GNN.

**Strengths:**

* The paper provides a fresh perspective on integrating homophily and heterophily in graph learning tasks;
* The presentation is easy to follow;
* Some experiments covering various datasets with different levels of homophily show significant improvements over state-of-the-art models.

**Weaknesses:**

* The paper does not address potential computational overhead introduced by the triple filter ensemble mechanism;
* The descriptions of the theorems are too lengthy and not concise.

**Questions:**

* Can the ensemble methods (EM1, EM2, EM3) be further optimized or replaced with alternative techniques to improve performance?
* What is the computational overhead of the triple filter ensemble mechanism compared to traditional spectral GNNs? How does the model scale with larger graphs and higher-dimensional feature spaces?
* The authors mention that TFE-GNN has higher generalization. Why? Any theoretical results for charactering the generalization ability of TFE-GNN?
* (I am quite curious about it) Are there specific scenarios or types of graphs where TFE-GNN might not perform as well?
*Authors may consider discussing the key technical differences between the proposed work and recent methods on heterophilous graph learning, such as: [1] Permutaion equivariant graph framelets for heterophilous graph learning; [2] How universal polynomial bases enhance spectral graph neural networks: Heterophily, over-smoothing, and over-squashing; [3] Flow2GNN: Flexible two-way flow message passing for enhancing GNNs beyond homophily

**Limitations:**

The authors have discussed the limitations of their work adequately.

---

> ### Author Rebuttal · Authors · 2024-08-07
>
> **We are grateful for your comments and suggestions, and we will explain and answer your concerns point by point!.**
>
> > **Weakness 1:** The potential computational overhead introduced by the triple filter ensemble.
>
> **W-A1:** Thanks for pointing out what we need to strengthen. The triple filter ensemble (TFE) is the core component of our proposed TFE-GNN, which combines filters to form a graph convolution. Similarly to ChebNetII [4] and PCNet [5], the time complexity of TFE-GNN is linear to $K_{lp}$+$K_{hp}$ when $EM_1$, $EM_2$, and $EM_3$ take summation. This shows that TFE does not influence the time complexity magnitude of the model.
>
> > **Weakness 2:** The descriptions of the theorems are too lengthy and not concise.
>
> **W-A2:** Thanks for your valuable and helpful comment. We simplify condition (2) of Theorem 1 as "$EM_1$, $EM_2$, and $EM_3$ take summation, $H_{hp}$ take $L_{sym}$ and $H_{lp}$ take $L_{sym}^a$". We simplify condition (2) of Theorem 2 as "$EM_1$, $EM_2$ takes summation and $EM_3$ takes ensemble method capable of preserving the model's properties, $H_{hp}$ takes $\bar{H}^2_{gf}$ and $H_{lp}$ takes $\bar{H}^1_{gf}$".
>
> > **Question 1:** Can the ensemble methods be further optimized or replaced?
>
> **Q-A1:** This is a insightful question that helps us analyze the TFE-GNN. There exist some other ensemble methods or alternative techniques to replace $EM_1$, $EM_2$, and $EM_3$, such as mean, maximization, pooling, etc., however, these methods do not perform as good as summation according to our experimental results.
>
> > **Question 2:** What is the computational overhead of the triple filter ensemble? How does the model scale with larger graphs and higher-dimensional feature spaces?
>
> **Q-A2:** Thanks for your insightful questions. Similar to most spectral GNNs [4, 5], the triple filter ensemble mechanism does not influence the time complexity magnitude of the TFE-GNN, i.e., the time complexity of TFE-GNN is linear to $K_{lp}$+$K_{hp}$ when $EM_1$, $EM_2$, and $EM_3$ take summation. Specifically, the time complexity of message propagation is $O((K_{lp}+K_{hp})|E|C)$, the time complexity of the combination of $H_{gf}$ with respectively $\omega$ and $\omega'$ (Equations 9 and 10) is $O((K_{lp}+K_{hp})nC)$, and the time complexity of the coefficient calculation is not greater than $O(K_{lp}+K_{hp})$, where $C$ denote the number of classes and $n$ denotes the number of nodes. We report the training time overhead of the different spectral GNNs in Table 6 of Appendix B.5.
>
> For graphs with higher dimensional features, we scale TFE-GNN by exchanging the order of message propagation and feature dimensionality reduction:
> $\widetilde{Z}=EM_3\lbrace\vartheta_1Z_{lp}, \vartheta_2Z_{hp}\rbrace = EM_3\lbrace \vartheta_1TFE_1f_{mlp}(X), \vartheta_2TFE_2f_{mlp}(X) \rbrace$. We use a sparse form of the adjacency matrix of large graphs, which greatly reduces the space required for TFE-GNN. Therefore, TFE-GNN scales well to large graphs and high-dimensional feature spaces.
>
> We chose two large graphs [6] --fb100-Penn94 and genius-- to verify the scalability of TFE-GNN, and the experimental results are 84.76\% and 89.32\%, respectively. **The performance of TFE-GNN on fb100-Penn94 exceeds that of all models mentioned in the article [6].** TFE-GNN tops the performance on genius dataset, which demonstrates its high competitiveness. All dataset statistics and experimental parameters are reported in Tables 1 and 2, respectively, of the author rebuttal’s one-page PDF. We will add a new section 3.5 to analyze and study the scalability of TFE-GNN in the final version.
>
> > **Question 3:** Why TFE-GNN has higher generalization.
>
> **Q-A3:** Thanks for your valuable and meaningful questions. TFE-GNN is free from polynomial computation, coefficient constraints, and specific scenarios. These factors, especially the coefficient constraints, are responsible for model overfitting [7]. We illustrate the generalization ability of TFE-GNN with more intuitive experiments. In addition to the two large graphs --fb100-Penn94 and genius--, we add two additional datasets [9] --roman-empire and amazon-rating-- whose edge homophily are 0.05 and 0.38 respectively, to verify the generalization of TFE-GNN. Experimental results on the two datasets are 75.87\%, and 52.21\%, respectively. These additional experimental results further validate the generalization ability of TFE-GNN on both homophilic and heterophilic datasets: **TFE-GNN can generalize well on graphs with different edge homophily levels.** We will follow up with sufficient experiments and report the experimental results and parameters in the final version.

---

### Official Review · Reviewer_pzPy · 2024-07-13

**Soundness:** 3
**Presentation:** 3
**Contribution:** 2
**Rating:** 5
**Confidence:** 4

**Summary:**

This paper proposes a spectral GNN with triple filter ensemble (TFE-GNN). As the ensemble of two diverse and simple low-pass and high-pass graph filters, TFE-GNN achieves good generalization ability and SOTA performance on both homophilic and heterophilic datasets.

**Strengths:**

S1. This work is motivated reasonably by leveraging the diversity of graph filters to create an ensemble GNN. It is well-explained with clear definitions, theories, and results. The results are further presented with visual tools to create an easy-to-understand intuition.

S2. The conclusions are supported by extensive experiments with a variety of baseline models and datasets.

**Weaknesses:**

W1. One major statement that TFE-GNN has better generalization then other adaptive filter GNNs is not fully explained - only empirical results on the loss curves are shown. As a major contribution emphasized in the abstract, more theoretical and empirical results are usually expected.

W2. The relevance of the abstract and introduction and the later sections can be enhanced. For example, most theoretical analyses are about the expressiveness of TFE-GNN, which is not stressed in the introduction.

**Questions:**

Questions:

Q1. In the introduction section, could you provide some references for the second problem, that the existing methods for solving problem one reduce the generalization ability of GNNs?

Q2. In Table 5 in Appendix A.2, is there a typo (93.13±0.62) in the last row of the second column? It significantly outperforms all other settings but is not marked in bold as the best result.

Q3. In Figure 5 in Appendix B.3, there is a large oscillation of the validation loss of the TFE-GNN, compared to other models. Could you account for this oscillation?



Suggestions:

S1. The idea of combining low-pass and high-pass filters is explored by many works (PCNet[1], GBK-GNN[2]). It would be interesting to explore the generalization ability between this class of two-fold filters versus general polynomial filters.

S2. The statement of the second problem (starting from line 64) that “Carefully crafted graph learning methods, sophisticated polynomial approximations, and refined coefficient constraints led to overfitting” needs clarification or references.


[1] Li, Bingheng, Erlin Pan, and Zhao Kang. "Pc-conv: Unifying homophily and heterophily with two-fold filtering." Proceedings of the AAAI Conference on Artificial Intelligence. Vol. 38. No. 12. 2024.

[2] Du, Lun, et al. "Gbk-gnn: Gated bi-kernel graph neural networks for modeling both homophily and heterophily." Proceedings of the ACM Web Conference 2022. 2022.

**Limitations:**

-

---

> ### Author Rebuttal · Authors · 2024-08-07
>
> **We sincerely appreciate your valuable comments and suggestions and we answer your concerns point to point!**
>
> > **Weakness 1:** More theoretical and empirical results are expected to explain why TFE-GNN has better generalization.
>
> **W-A1:** Thanks for your meaningful comments. We further explain why TFE-GNN has better generalization by introducing two new references [5, 6]. FFKSF [5] and ARMA [6] attribute the model overfitting to the overfitting of polynomial coefficients and the aggregation of high-order neighborhood information. TFE-GNN does not impose refined constraints on the coefficients and achieves better performance with a smaller order, whcih possesses higher generalization.
>
> We select four new datasets [2, 3] and conduct relevant experiments to further explain why TFE-GNN has better generalization, including roman-empire, amazon-rating, and two large graphs [3] fb100-Penn94 and genius, whose edge homophily are 0.05, 0.38, 0.47, and 0.62, respectively. Experimental results show that TFE-GNN achieves competitive performance on these datasets with results of 75.87\%, 52.21\%, 84.76\% and 89.32\%, respectively. TFE-GNN achieves competitive rankings on all datasets, with the best performance on fb100-Penn94, outperforming most spectral GNNs on roman-empire, and topping the rest of the datasets. These additional experimental results further validate the generalization ability of TFE-GNN on both homophilic and heterophilic datasets: **TFE-GNN can generalize well on graphs with different edge homophily levels.** We will add the above relevant content to Section 4.3 and Appendix B.3 in the final version.
>
> > **Weakness 2:** The relevance of the abstract and introduction and the later sections can be enhanced.
>
> **W-A2:** Thanks for pointing out our current shortcomings. To stress the expressiveness of TFE-GNN, we add the following sentences from line 83 of the paper: Theorem 2 show that TFE-GNN is a reasonable combination of two excellent polynomial-based spectral GNNs, which motivates it to perform well. TFE-GNN preserves the ability of the different filters while reducing overfitting problem.
>
> > **Question 1:** Could you provide some references for the second problem?
>
> **Q-A1:** We appreciate your valuable question and find that the second problem, which builds on the first, is really not sufficiently described. We add the following sentences to clarify it from line 67 of the introduction: FFKSF [5] attributes the degradation of polynomial filters’ performance to the overfitting of polynomial coefficients. ChebNetII [4] further constrains the coefficients to enable them easier to be optimized. ARMA [6] suggests that the filter will overfit the training data when aggregating high-order neighbor information. Whereas the order of polynomial-based spectral GNNs is usually large to increase the approximation of the polynomials, which directs them to obtain high-order neighborhood information, and then leads to overfitting. Therefore, it is reasonable to assume that carefully crafted graph learning methods, sophisticated polynomial approximations, and refined coefficient constraints lead to overfitting of the models, which diminishes generalization of the models. TFE-GNN discards these factors to improve generalization, while retaining the approximation ability.
>
> > **Question 2:** In Table 5 in Appendix A.2, is there a typo (93.13±0.62) in the last row of the second column?
>
> **Q-A2:** Thanks for your responsible, careful and valuable comment! We are sorry for our carelessness. Actually, its correct value should be 83.13±0.62, we will revise it in the final version.
>
> > **Question 3:** Could you account for this oscillation in Figure 5 in Appendix B.3?
>
> **Q-A3:** Thanks for your valuable comment. The learning rate controls the step size which in turn affects the loss optimization. The large learning rate (= 0.1) is responsible for the oscillations of the validation loss in Figure 5 in Appendix B.3. Figure 1 in the one-page PDF in the author rebuttal shows that the loss is stable when the learning rate is 0.001. The early stopping mechanism allows TFE-GNN to carry less stable losses and losses do not fall into unacceptable local maximum.
>
> > **Suggestion 1:** It would be interesting to explore the generalization ability between this class of two-fold filters versus general polynomial filters.
>
> **S-A1:** Thanks for your valuable ideas. We will add the following to Section 5: We are eager to explore different types of combinations of low-pass and high-pass filters, as well as global combining methods such as pooling.
>
> > **Suggestion 2:** The statement of the second problem needs clarification or references.
>
> **S-A2:** Thank you for your helpful suggestion. In addition to PCNet [1] pointing out that carefully crafted graph learning methods lead to low model generalization, we add two new references [5, 6] to state the second problem of this paper. FFKSF [5] and ARMA [6] attribute the overfitting to the overfitting of polynomial coefficients and the aggregation of high-order neighborhood information. We add the relevant analysis to the introduction starting at line 67 of the introduction
>
> We appreciate the advice! We will gladly answer any additional questions you may have.
>
> **References**
>
> [1] Li, B et al. Pc-conv: Unifying homophily and heterophily with two-fold filtering. AAAI 2024.
>
> [2] Oleg Platonov et al. A critical look at the evaluation of GNNs under heterophily: Are we really making progress?. ICLR 2022.
>
> [3] Derek Lim et al. Large scale learning on non-homophilous graphs: New benchmarks and strong simple methods. NeurIPS 2021.
>
> [4] He, M et al. Convolutional neural networks on graphs with chebyshev approximation, revisited. NeurIPS 2022.
>
> [5] Z. Zeng et al. Graph Neural Networks With High-Order Polynomial Spectral Filters. IEEE TNNLS, 2023.
>
> [6] F. M. Bianchi et al.. Alippi, Graph Neural Networks With Convolutional ARMA Filters. IEEE TPAMI, 2022.

---

> > ### Comment · Reviewer_pzPy · 2024-08-13
> > **Response to the rebuttal**
> >
> > Thank you for your detailed explanations to address the concerns and questions. Most of my questions are properly explained. The experiments are also extended to provide more solid proof of generalization: The authors introduced datasets of different edge homophily levels to explain how TFE-GNN generalize on diverse datasets. A theoretical explanation is also provided based on some references. I am expecting a more in-depth explanation on role of the key innovation - ensembleod nonetheless.
> >
> > This paper introduces a spectral GNN with triple filter ensemble. Through experiment results are theoretical explanations, the authors proved that TFE-GNN has strong approximation ability and generalization ability. This work provides an interesting combination of the graph filters and ensemble methods to produce a robust graph learning method.
> >
> > Based on the contributions and results of the work, I consider it acceptable to the conference. I hope the authors can delve into the origin of the effectiveness for a better understanding of the GNN community.

---

> ### Author Response · Authors · 2024-08-14
> **Response to Reviewer pzPy**
>
> Thanks for the recognition and insightful comments. We will answer your concerns below.
>
> > more in-depth explanation on role of the key innovation – ensemble and the origin of the effectiveness.
>
> We explain in depth "the ensemble and the origins of effectiveness" in terms of the problems solved by TFE-GNN. Then, we explain how TFE-GNN accomplishes our purpose and explain what the generalization and performance of TFE-GNN stems from.  Finally, we elaborate on the key differences between TFE-GNN and filter combinations [4, 5], filter banks [6] and other similar methods [7. 8].
>
> + We describe the core problem in this paper as follows: Carefully crafted graph learning methods, sophisticated polynomial approximations, and refined coefficient constraints leaded to overfitting, which diminishes GNNs’ generalization. We are inspired by the properties of ensemble learning [1, 2, 3] and use ensemble ideas to solve this problem. First, the strong classifier determined by the base classifiers can be more accurate than any of them if the base classifiers are accurate and diverse. And then, this strong classifier retains the characteristics of the base classifier to some extent.
>
> + **Our TFE-GNN utilizes the above properties of ensemble to achieve the following effects: improves generalization and hence classification performance while retaining approximation ability.** Specifically, we combine a set of weak base low-pass filter to determine a strong low-pass filter that can extract homophily from graphs. We combine a set of weak base high-pass filter to determine a strong high-pass filter that can extract heterophily from graphs. Learnable coefficients of low-pass/high-pass filters can enhance the adaptivity of TFE-GNNs. Finally, TFE-Conv is generated by combining the above two strong filters with two learnable coefficients, which retains the characteristics of both two strong filters, i.e., it can extract homophily and heterophily from graphs adaptively. Furthermore, Theorems 1 and 2 prove the approximation ability and expressive power of TFE-GNN, and experimental results verify that TFE-GNN achieves our purpose.  Therefore, the high generalization and state-of-the-art classification performance of TFE-GNN stems from the aptness of the base classifiers and the advantages of the ensemble idea.
>
> + The key difference between TFE-GNN and other GNNs with similar aggregation paradigms [4-8] is that **TFE-GNN retains the ability of polynomial-based spectral GNNs while getting rid of polynomial computations, coefficient constraints, and specific scenarios.** Specifically, Equation 13 and its predecessors in our paper show that TFE-GNN is free from polynomial computations, unlike ChebNetII and PCNet which require the computation of Chebyshev polynomials $T_k(x)$(Eq. 8 in [9]) and Possion-Charlier polynomials $C_n(k, t)$ (Eq. 15 in [4]), respectively. TFE-GNN does not impose refined constraints on the coefficients and does not design very complex learning methods, which helps it avoid overfitting as much as possible. TFE-GNN extracts homophily and heterophily from graphs with different levels of homophily adaptively while utilizing the initial features, which helps it not be limited to specific scenarios. In summary, TFE-GNN achieves a better trade-off in approximation ability and classification performance.
>
> We will add relevant content to the Introduction and 3.1 Motivation of the final version and cite relevant references, including suggested relevant references. Thanks again for your valuable and insightful comments.
>
> **References**
>
> [1] Schapire, R. E. The strength of weak learnability. Machine learning, 5(2):197–227, 1990.
>
> [2] Hansen, L. K. and Salamon, P. Neural network ensembles. IEEE transactions on pattern analysis and machine intelligence, 12(10):993–1001, 1990.
>
> [3] Zhou, Z.-H. Ensemble methods: foundations and algorithms. 2012.
>
> [4] Li, B et al. Pc-conv: Unifying homophily and heterophily with two-fold filtering. AAAI 2024.
>
> [5] Du, Lun, et al. Gbk-gnn: Gated bi-kernel graph neural networks for modeling both homophily and heterophily. Proceedings of the ACM Web Conference 2022. 2022.
>
> [6] Sitao Luan, et al. Revisiting heterophily for graph neural networks. NeurIPS 2022.
>
> [7] Keke Huang et al. How universal polynomial bases enhance spectral graph neural networks: Heterophily, over-smoothing, and over-squashing. ICML, 2024.
>
> [8] Changqin Huang et al. Flow2GNN: Flexible two-way flow message passing for enhancing GNNs beyond homophily. IEEE TRANSACTIONS ON CYBERNETICS, 2024.
>
> [9] He, M et al. Convolutional neural networks on graphs with chebyshev approximation, revisited. NeurIPS 2022.

---

### Official Review · Reviewer_eCxX · 2024-07-14

**Soundness:** 3
**Presentation:** 3
**Contribution:** 3
**Rating:** 6
**Confidence:** 4

**Summary:**

This paper proposed a spectral GNN with triple filter ensemble (TFE-GNN) that aims to retain both the ability of polynomial based spectral GNNs to approximate filters and the higher generalization and performance on graph learning tasks, when most existing GNNs can only retain one of the two capabilities. Theoretical analysis shows that the approximation ability of TFE-GNN is consistent with that of ChebNet under certain conditions, namely it can learn arbitrary filters. Experiments show that TFE-GNN achieves high generalization and best performance on various homophilous and heterophilous real-world datasets.

**Strengths:**

- The experiments are thoroughly conducted with 10 datasets and 17 baselines, which show that the proposed TFE-GNN achieve the best performance over all baseline models.
- The introduction of learnable spectral GNNs and the proposed TFE-GNN is clear and in details.

**Weaknesses:**

- My main concern is that the heterophilous datasets that are used in the experiments are not up-to-date. Specifically, the five heterophilous datasets in the experiments are a subset of those first adopted by Pei et al. 2020 (Geom-gcn).  However, recent works like Platonov et al. 2022 has identified several drawbacks of these datasets, including leakage of test nodes in the training set in squirrel and chameleon, which can render the results obtained on these datasets unreliable. I recommend the authors to run additional experiments on two more up-to-date heterophilous datasets, such as those introduced in the papers below:
    - Oleg Platonov, Denis Kuznedelev, Michael Diskin, Artem Babenko, and Liudmila Prokhorenkova. 2022. A critical look at the evaluation of GNNs under heterophily: Are we really making progress?. In The Eleventh International Conference on Learning Representations.
    - Derek Lim, Felix Hohne, Xiuyu Li, Sijia Linda Huang, Vaishnavi Gupta, Omkar Bhalerao,
    and Ser Nam Lim. 2021. Large scale learning on non-homophilous graphs: New benchmarks and strong simple methods. Advances in Neural Information Processing Systems 34 (2021), 20887–20902.
- The authors discussed that for existing GNN models, “some polynomial-based models use polynomials with better approximation than some other models when approximating filters, but the former’s performance is lagging behind that of the latter on real-world graphs.” The proposed TFE-GNN models seem to have overcome that limitation, but it remains unclear to me in a high-level, what are the key differences between TFE-GNN compared to prior models which make TFE-GNN able to achieve a better trade-off here?

**Questions:**

It would be great if the authors can address the points discussed in weakness during the rebuttal.

**Limitations:**

The paper includes discussion regarding the limitations in Appendix A.2, most notably the reliance on setting hyperparameters $K_{lp}$ and $K_{hp}$ that are suitable for the homophily level of datasets.

---

> ### Author Rebuttal · Authors · 2024-08-07
>
> **We appreciate your valuable suggestions and we will explain them line by line.**
>
> > **Question 1:** I recommend the authors to run additional experiments on two more up-to-date heterophilous datasets, such as those introduced in the papers  [1, 2].
>
> **A1:** Thank you for your valuable comment. We think that adding relevant experiments is necessary to make our paper more complete after discussing it. We validate our model with additional experiments on two large graphs [2] --fb100-Penn94 and genius-- whose edge homophily are 0.47 and 0.62, respectively. Experimental results on these two datasets were 84.76\% and 89.32\% for node classification. **The performance of TFE-GNN on fb100-Penn94 dataset exceeds that of all models mentioned in the article [2].** TFE-GNN tops the performance on genius dataset, which demonstrates its high competitiveness. All dataset statistics and preliminary experimental parameters are reported in Tables 1 and 2, respectively, of the author rebuttal’s one-page PDF. Note that we use the same dataset splits as in the article [2].
>
> We will add relevant experiments and their results analysis to the final version. We will modify our publicly available code accordingly and follow up with sufficient experiments and report the experimental results and parameters in the final version.
>
> > **Question 2:** It remains unclear to me in a high-level, what are the key differences between TFE-GNN compared to prior models which make TFE-GNN able to achieve a better trade-off here?
>
> **A2:** Thanks for your constructive feedback. The key difference between TFE-GNN and prior models is that **TFE-GNN retains the ability of polynomial-based spectral GNNs while getting rid of polynomial computations, coefficient constraints, and specific scenarios.** The first problem we describe in line 52 of the introduction shows that it is not only the approximation ability of the polynomials affects the performance of GNNs, but also coefficient constraints and graph learning methods influence the performance. For example, ChebNetII [3], a prior model, constrains learnable coefficients by Chebyshev interpolation to boost ChebNet’s performance, which brings overfitting [6] for it. EvenNet [4] and PCNet [5] elaborate graph learning methods, namely even-polynomials and heterophilic graph heat kernel to ignore odd-hop neighbors, whcih makes them to accommodate the structure nature of heterophilic graph. This elaborate design is similar to the heterophily-specific models [7]. These models achieve great node classification performance for specific problems or scenarios through specific tricks.
>
> In this paper, our proposed model is free from polynomial computations, coefficient constraints, and specific scenarios. Specifically, Equation 13 and its predecessors in our paper show that TFE-GNN is free from polynomial computations, unlike ChebNetII and PCNet which require the computation of Chebyshev polynomials $T_k(x)$(Eq. 8 in [3]) and Possion-Charlier polynomials $C_n(k, t)$ (Eq. 15 in [5]), respectively. TFE-GNN does not impose refined constraints on the coefficients and does not design very complex learning methods, which helps it avoid overfitting as much as possible. TFE-GNN extracts homophily and heterophily from graphs with different levels of homophily adaptively while utilizing the initial features, which helps it not be limited to specific scenarios. Furthermore, Theorems 1 and 2 prove the approximation ability and expressive power of TFE-GNN, and experimental results verify that TFE-GNN achieves our purpose. In summary, TFE-GNN achieves a better trade-off in approximation ability and classification performance.
>
> We will add the following sentences from line 80 of the introduction: The key difference between TFE-GNN and prior models is that TFE-GNN retains the ability of polynomial-based spectral GNNs while getting rid of polynomial computations, coefficient constraints, and specific scenarios.
>
> We appreciate the advice! We will gladly answer any additional questions you may have.
>
> **References**
>
> [1] Oleg Platonov et al. A critical look at the evaluation of GNNs under heterophily: Are we really making progress?. ICLR 2022.
>
> [2] Derek Lim et al. Large scale learning on non-homophilous graphs: New benchmarks and strong simple methods. NeurIPS 2021.
>
> [3] He, M et al. Convolutional neural networks on graphs with chebyshev approximation, revisited. NeurIPS 2022.
>
> [4] Lei, R et al. Evennet: Ignoring odd-hop neighbors improves robustness of graph neural networks. NeurIPS 2021.
>
> [5] Li, B et al. Pc-conv: Unifying homophily and heterophily with two-fold filtering. AAAI 2024.
>
> [6] Z. Zeng et al. Graph Neural Networks With High-Order Polynomial Spectral Filters. IEEE Transactions on Neural Networks and Learning Systems, 2023.
>
> [7] Oleg Platonov et al. A critical look at the evaluation of GNNs under heterophily: Are we really making progress? ICLR 2022.

---

> ### Comment · Reviewer_eCxX · 2024-08-13
> **Response to Authors' Rebuttal**
>
> I appreciate the authors detailed response to my comments. After reading authors' response, I decide to keep my original rating.

---

> > ### Author Response · Authors · 2024-08-13
> > **Response to Reviewer eCxX**
> >
> > We thank the reviewer for reading and agreeing with our responses, and we appreciate that the reviewer agreed with our paper and kept the original rating. We will add the relevant parts of the response to the final version.

---

### Author Rebuttal · Authors · 2024-08-07

**Global Response**

We sincerely thank all reviewers for their thoughtful and constructive feedback and their recognition of our work! We have made significant improvements to our initial results in response to their comments. We add a new figure and two tables in the one-page PDF and update the paper and supplementary PDFs with revisions accordingly. We welcome any follow-up discussions!

**Time Complexity and Scalability.** We further investigate the time complexity of TFE-GNN and add relevant results to Appendix B.5 in the final version. Similarly to ChebNetII [1] and PCNet [2], the time complexity of TFE-GNN is linear to $K_{lp}$+$K_{hp}$ when $EM_1$, $EM_2$, and $EM_3$ take summation. Specifically, the time complexity of message propagation is $O((K_{lp}+K_{hp})|E|C)$, the time complexity of the combination of $\omega$ and $H_{gf}$ (Equations 9 and 10) is $O((K_{lp}+K_{hp})nC)$, and the time complexity of the coefficient calculation is not greater than $O((K_{lp}+K_{hp}))$, where $C$ denote the number of classes and $n$ denotes the number of nodes.

For graphs with higher dimensional features, we scale TFE-GNN by exchanging the order of message propagation and feature dimensionality reduction:
$\widetilde{Z}=EM_3\lbrace\vartheta_1Z_{lp}, \vartheta_2Z_{hp}\rbrace = EM_3\lbrace \vartheta_1TFE_1f_{mlp}(X), \vartheta_2TFE_2f_{mlp}(X) \rbrace$. We use a sparse representation of the adjacency matrix of large graphs, which greatly reduces the space required for TFE-GNN. The new form and the time complexity of TFE-GNN show that **TFE-GNN scales well to large graphs and high-dimensional feature spaces.**

**Generalization Analysis.** We further illustrate the generalization ability of TFE-GNN by introducing new references. PCNet [2] points out that many carefully crafted graph representation learning methods are incapable of generalizing well across real-world graphs with different levels of homophily. FFKSF [3] attributes the degradation of polynomial filters’ performance to the overfitting of polynomial coefficients, and some spectral GNNs, such as ChebNetII [1], further constrains the coefficients to enable them easier to be optimize. ARMA [6] suggests that the filter will overfit the training data when aggregating high-order neighbor information. Whereas the order of polynomial-based spectral GNNs is usually large to increase the approximation of the polynomials, which directs them to obtain high-order neighborhood information, which in turn leads to overfitting. Therefore, it is reasonable to assume that carefully crafted graph learning methods, sophisticated polynomial approximations, and refined coefficient constraints lead to overfitting of the models, which diminishes generalization of the models. TFE-GNN discards these factors to improve generalization, while retaining the approximation ability.

**More Experiments.** We conduct more experiments for verifying the performance, generalization and scalability of TFE-GNN on four additional datasets. We select two heterophilic [4] --roman-empire and amazon-rating-- whose edge homophily are 0.05 and 0.38 respectively, and select two large graphs [5] --fb100-Penn94 and genius-- whose edge homophily are 0.47 and 0.62 respectively. Note that we use the same dataset splits as the provenance of these datasets. Experiments results on these datasets are 75.87\%, 52.21\%, 84.76\% and 89.32\%, respectively. TFE-GNN achieves competitive rankings on all datasets, with the best performance on fb100-Penn94, outperforming most spectral GNNs on roman-empire, and topping the rest of the datasets. It is worth noting that TFE outperforms all SOTA models in the article [4] on homophilic datasets. These additional experimental results further validate the generalization ability of TFE-GNN on both homophilic and heterophilic datasets: **TFE-GNN can generalize well on graphs with different edge homophily levels.** All dataset statistics and experimental parameters are reported in Tables 1 and 2, respectively, of the one-page PDF. The very competitive results shown by TFE-GNN on these datasets demonstrate its good generalization ability and scalability.

**Loss Oscillation.** The learning rate controls the step size which in turn affects the loss optimization. The large learning rate (= 0.1) is responsible for the oscillations of the validation loss in Figure 5 in Appendix B.3. Figure 1 in the one-page PDF shows that the loss is stable  when the learning rate is 0.001.

**TFE-GNN Versus Other GNNs with Similar Aggregation Paradigms.** The key technical differences: our proposed TFE-GNN is free from polynomial computation, coefficient constraints, and specific scenarios, compared to polynomial-based spectral GNNs and heterophily-specific Models [4], such as ChebNetII [1] and PCNet [2]. TFE-GNN achieves higher performance through a simpler network structure compared to PEGFAN [7] and UniFilter[8].

**References**

[1] He, M et.al. Convolutional neural networks on graphs with chebyshev approximation, revisited. NeurIPS, 2022.

[2] Li, B et.al. Pc-conv: Unifying homophily and heterophily with two-fold filtering. AAAI, 2024.

[3] Z. Zeng et al. Graph Neural Networks With High-Order Polynomial Spectral Filters. IEEE Transactions on Neural Networks and Learning Systems, 2023.

[4] Oleg Platonov et al. A critical look at the evaluation of GNNs under heterophily: Are we really making progress?. ICLR 2022.

[5] Derek Lim et al. Large scale learning on non-homophilous graphs: New benchmarks and strong simple methods. NeurIPS 2021.

[6] F. M. Bianchi et al.. Alippi, Graph Neural Networks With Convolutional ARMA Filters. IEEE TPAMI, 2022.

[7] Jianfei Li et al. Permutaion equivariant graph framelets for heterophilous graph learning. TNNLS, 2024.

[8] Keke Huang et al. How universal polynomial bases enhance spectral graph neural networks: Heterophily, over-smoothing, and over-squashing. ICML, 2024.

---

### Author Response · Authors · 2024-08-07
**Explanation of Question 4 on Reviewer zzRJ**

> **Question 4:** (I am quite curious about it) Are there specific scenarios or types of graphs where TFE-GNN might not perform as well? The key technical differences between the proposed work and recent methods on heterophilous graph learning.

**Q-A4:** Thanks for your valuable and constructive questions. TFE-GNN is more suitable for full-supervised node classification than semi-supervised node classification. **The key technical differences: TFE-GNN is free from polynomial computation, coefficient constraints, and specific scenarios,** compared to polynomial-based spectral GNNs and heterophily-specific Models [8], such as ChebNetII [4] and PCNet [5]. The key differences between TFE-GNN and the models mentioned in this question are the following.

+ Flow2GNN [3] is an interesting and novel attempt, which decomposes the original adjacency matrix into two matrices via a random binary matrix $G^{(l)}$ with elements that follow the Bernoulli distribution. Information flows within the nodes in two respective disentangled graphs with reduced heterophily, and then adaptively aggregates them with the strength estimation vector $p_i^{(l)}$ of information flow. **TFE-GNN combines the well-performing and well-tested graph filters, while Flow2GNN is highly dependent on $G^{(l)}$ and $p_i^{(l)}$.**

+ UniFilter [2] establishes a link between the estimated graph homophily rate $\hat{h}$ and the propagation matrix $P$ by $\theta =\pi /2(1-\hat{h})$, and forms heterophily bases $\lbrace u_0,u_1,…,u_K \rbrace$ according to Algorithm1 and then obtain the rotation matrix $P_{\theta}$. UniFilter combines $P$ and $P_{\theta}$ using the coefficient $\tau$ and its weight vector $w$
discloses the significance of each frequency component in graph. The performance of UniFilter is affected by the $\hat{h}$ estimated from the training data labels, which seems to require that the training data label distribution is similar to the graph true global label distribution. **The difference between TFE-GNN and UniFilter is that TFE-GNN rarely relies on such priori knowledge and conditions, which ensures its better adaptability and classification performance.**

+ PEGFAN [1] sets up three optional feature matrices for the homophilic and heterophilic graphs, respectively, and performs a concatenation operation for the selected feature matrices. PEGFAN performs row normalization on the feature matrices that message passing and dimensionality-reducing outputs, and adds a new linear layer before $softmax$ after the activation function $ReLU$. **These components increase network complexity of PEGFAN, whereas TFE-GNN is more concise and requires less space.**

+ We will add a new section "TFE-GNN Versus Other GNNs with Similar Aggregation Paradigms", named B.6, to analyze these models and add the above related content to the appendix of the final version.

**References**

[1] Jianfei Li et al. Permutaion equivariant graph framelets for heterophilous graph learning. TNNLS, 2024.

[2] Keke Huang et al. How universal polynomial bases enhance spectral graph neural networks: Heterophily, over-smoothing, and over-squashing. ICML, 2024.

[3] Changqin Huang et al. Flow2GNN: Flexible two-way flow message passing for enhancing GNNs beyond homophily. IEEE TRANSACTIONS ON CYBERNETICS, 2024.

[4] He, M et.al. Convolutional neural networks on graphs with chebyshev approximation, revisited. NeurIPS, 2022.

[5] Li, B et.al. Pc-conv: Unifying homophily and heterophily with two-fold filtering. AAAI, 2024.

[6] Derek Lim et.al. Large scale learning on non-homophilous graphs: New benchmarks and strong simple methods. NeurIPS, 2021.

[7] Z. Zeng et.al. Graph Neural Networks With High-Order Polynomial Spectral Filters. IEEE TNNLS, 2023.

[8] Oleg Platonov et.al. A critical look at the evaluation of GNNs under heterophily: Are we really making progress? ICLR 2022.

---

> ### Comment · Reviewer_zzRJ · 2024-08-13
>
> Thank you for the detailed rebuttal. I am satisfied with the responses provided. However, I recommend that the authors properly cite the suggested references on heterophilous graph learning in the final version. Based on this, I will raise my score to 6.

---

> > ### Author Response · Authors · 2024-08-13
> > **Response to Reviewer zzRJ**
> >
> > Thanks for your helpful feedback! We will cite the suggested references in the final version because of the addition of a new section entitled "TFE-GNN Versus Other GNNs with Similar Aggregation Paradigms" and add the relevant parts of the response to the final version.

---

### Decision · Program_Chairs · 2024-09-25

**Decision:**

Accept (poster)

**Comment:**

In this paper, the authors proposed a new GNN model based on a triple filter ensemble mechanism (TFE-GNN), which aims to learn a spectral GNN with theoretically guaranteed filter approximation power and strong generalizability in practical node classification tasks. In theory, the authors demonstrated that the proposed TFE-GNN is equivalent to ChebNet under some specific settings, so they have the same filter approximation power. Moreover, this model leads to the combination of two spectral GNNs that correspond to heterophilic and homophilic graphs, respectively, which provides a unified GNN framework. Experiments show that the proposed method achieves promising performance in various node classification tasks.

All three reviewers scored this paper positively, and their concerns about this work's motivation, complexity, and solidness were resolved in the rebuttal phase. In addition, although both SAC and I agree that the merits of this paper seem to outweigh the issues, we believe that more generalizability tests are required, e.g., testing model performance with increased data noise or in some OOD scenarios, and that comparisons with baselines like ChebNet in such scenarios are necessary. Otherwise, the claims in the abstract and introduction are not convincing to some extent.

In summary, I decided to accept this work and strongly suggest the authors polish this paper according to the reviewers' comments.